# Bid Protein: A Participant in the Apoptotic Network with Roles in Viral Infections

**DOI:** 10.3390/ijms26062385

**Published:** 2025-03-07

**Authors:** Zbigniew Wyżewski, Karolina Paulina Gregorczyk-Zboroch, Matylda Barbara Mielcarska, Weronika Świtlik, Adrianna Niedzielska

**Affiliations:** 1Institute of Biological Sciences, Cardinal Stefan Wyszynski University in Warsaw, Dewajtis 5, 01-815 Warsaw, Poland; 2Division of Immunology, Department of Preclinical Sciences, Institute of Veterinary Medicine, Warsaw University of Life Sciences—SGGW, Ciszewskiego 8, 02-786 Warsaw, Poland; karolina_gregorczyk-zboroch@sggw.edu.pl (K.P.G.-Z.); matylda_mielcarska@sggw.edu.pl (M.B.M.); adrianna_niedzielska@sggw.edu.pl (A.N.); 3Centre for Advanced Materials and Technologies, Warsaw University of Technology, Poleczki 19, 02-822 Warsaw, Poland; weronika.switlik@pw.edu.pl

**Keywords:** apoptosis, Bid, tBid, Bcl-2 family, mitochondria, HBV, HSV, IAV, SARS-CoV-2, HIV

## Abstract

The BH3-interacting domain death agonist (Bid), a proapoptotic signaling molecule of the B-cell lymphoma 2 (Bcl-2) family, is a key regulator of mitochondrial outer membrane (MOM) permeability. Uniquely positioned at the intersection of extrinsic and intrinsic apoptosis pathways, Bid links death receptor signaling to the mitochondria-dependent cascade and can also be activated by endoplasmic reticulum (ER) stress. In its active forms, cleaved Bid (cBid) and truncated Bid (tBid), it disrupts MOM integrity via Bax/Bak-dependent and independent mechanisms. Apoptosis plays a dual role in viral infections, either promoting or counteracting viral propagation. Consequently, viruses modulate Bid signaling to favor their replication. The deregulation of Bid activity contributes to oncogenic transformation, inflammation, immunosuppression, neurotoxicity, and pathogen propagation during various viral infections. In this work, we explore Bid’s structure, function, activation processes, and mitochondrial targeting. We describe its role in apoptosis induction and its involvement in infections with multiple viruses. Additionally, we discuss the therapeutic potential of Bid in antiviral strategies. Understanding Bid’s signaling pathways offers valuable insights into host–virus interactions and the pathogenesis of infections. This knowledge may facilitate the development of novel therapeutic approaches to combat virus-associated diseases effectively.

## 1. Introduction

The extremely high complexity of physiological processes within multicellular organisms necessitates strict molecular regulation. Both extracellular and intracellular signaling networks determine the homeostasis of the biological systems, allowing them to obtain long-term stability and maintain individual development. Morphogenesis and the proper functioning of the organisms demand the cooperation of particular cells, tissues, and organs. In a healthy organism, every single component should be subjected to the overarching structures and fit into the scenario of the whole body development. Therefore, the variety of molecular pathways and regulatory mechanisms tailor the viability of individual cells to the needs of the multicellular organism [1,2,3,4,5,6,7,8,9,10]. The old, damaged, malfunctioning, and infected ones undergo elimination via type I programmed cell death (PCD), also called apoptosis. This process is initiated in response to various extracellular and intracellular stimuli and regulated by miscellaneous molecular factors [9,11,12,13,14,15,16]. Mechanisms determining cell fate maintain the organism’s health, as the deregulation of apoptosis contributes to many pathological stages, including viral infections and oncogenesis [17,18,19,20,21].

Type I PCD can be part of the host’s antiviral strategy. By effectively eliminating infected cells, it may disrupt pathogen DNA/RNA replication and the production of progeny virions. On a systemic scale, it can prevent the spread of the “intruder” throughout the organism. However, distinct viruses can affect the viability of infected cells, promoting pathogen propagation, impairing the immune response, or establishing a latent stage that allows them to persist in the host while evading its defense mechanisms. Viral infections can either promote or suppress apoptosis, as viruses encode proteins that interfere with molecular signaling. The modulation of cell death may involve both the extrinsic and intrinsic pathways of type I PCD [17,18,19,20,21].

The BH3-interacting domain death agonist (Bid), a protein that bridges receptor-mediated and mitochondria-dependent apoptotic cascades, may play a role in various viral infections. Bid belongs to the B-cell lymphoma 2 (Bcl-2) family factors, a large group of apoptosis regulators [22]. Scientific research has linked various pathological stages to the changes in the level and the activity of different Bcl-2 family members [10,23,24,25,26,27,28]. This article focuses on Bid protein and its role in viral infections. We describe the mechanisms underpinning Bid-dependent apoptosis induction, Bid’s structural adaptations for its regulatory functions, its activation process, mitochondrial targeting, and the tBid-driven permeabilization of the mitochondrial outer membrane (MOM). We also discuss the importance of the Bid molecule in the context of the course and/or potential treatment of infections with several viruses, including the hepatitis B virus (HBV), herpes simplex virus type 2 (HSV-2), severe acute respiratory syndrome coronavirus 2 (SARS-CoV-2), influenza A virus (IAV), and human immunodeficiency virus (HIV). The pathogens selected for this review pose a significant challenge to modern medicine. They infect humans, resulting in a wide range of health consequences, from chronic diseases to severe acute outcomes. Due to their clinical significance and documented association with Bid-dependent apoptosis regulation, these pathogens were chosen for this article. A deeper understanding of Bid’s role in determining the fate of infected cells could aid in the search for effective antiviral treatment strategies.

## 2. Bid Protein—The Structure and the Role in Apoptosis Regulation

### 2.1. The Structure of Bid Protein

Bid protein is a 22 kDa molecule composed of eight α-helices, including six amphipathic (αH1-5 and αH8) and two hydrophobic ones (αH6 and αH7). αH6 and αH7 are localized in the protein interior and encircled by the remaining helices in a conformation that resembles the molecular architecture of transmembrane bacterial toxins (e.g., colicins). The hydrophobic hairpin structure constituted by αH6 and αH7 may embed in the MOM, anchoring the tBid molecule in the organelle [29,30].

The fragment, comprising nine amino acid residues at 90–98 positions (90–98 aa, within αH3), constitutes the Bcl-2 homology (BH)3 domain, a structural characteristic of BH3-only proteins from the Bcl-2 family of apoptosis regulators. The BH3 domain enables the active form of Bid to interact with other members of the Bcl-2 family, such as proapoptotic Bcl-2-associated X (Bax) [29].

Bid consists of an unstructured loop (42–79 aa) between αH2 and αH3 [29] (Billen). This region is a target structure for the proteolytic enzymes that cleave Bid protein, including caspase-8, granzyme B, calpain, and cathepsins [29,31,32,33]. Moreover, the unstructured loop comprises threonine residue at position 59 (Thr59) that may undergo phosphorylation by casein kinase 2 (CK2). This modification renders Bid protein resistant to the hydrolytic activity of the proteases [29,34].

Since unphosphorylated Bid is sensitive to proteolysis, two parts of the molecule can be distinguished as the products of the potential cleavage: the 7kDa (p7) N-terminal fragment and the 15 kDa (p15) C-terminal one, called truncated Bid (tBid). The latter is the active form of the protein [35]. Different reports indicate distinct places of proteolytic cleavage within the Bid molecule. According to the findings of Gahl et al. [36], the site of caspase-8-mediated splitting within the human Bid encoded by the pET15b-Bid plasmid may be the leucine residue at position 56 (Leu56) [36]. Additionally, an early in vitro experiment by Zha et al. [37] performed on the rabbit reticulocyte lysate identified glycine (Gly)60 as the N-terminal amino acid residue of tBid. In light of the findings mentioned above, Gly60, from the conserved Gly-Ser-Gln-Ala-Ser-Arg motif, is the site of myristoylation [37], a post-translational modification consisting of the attachment of the myristic acid (MyA) rest to the target structure [38]. The MyA is bound to Gly60 via the amide linkage, the formation of which requires a free amino group reconstructed by caspase-8-mediated proteolysis [39]. After the proteolytic cleavage, p7 and tBid may stay non-covalently bound. The myristoylation of tBid induces the structural rearrangement that favors the association of the p7-tBid complex to the MOM and promotes mitochondria-dependent apoptosis [37].

### 2.2. Bid Protein in the Network of Apoptosis Regulatory Molecules

The Bid molecule represents the Bcl-2 family proteins, a group of factors comprising both pro- and antiapoptotic members. They are responsible for the regulation of the intrinsic apoptosis pathway, affecting the integrity of the MOM and, thus, determining the intracellular localization of mitochondrial proapoptotic factors [40,41,42,43,44,45]. Two Bcl-2 family members, Bax and Bcl-2 homologous antagonist/killer (Bak), are able to oligomerize within MOM and constitute protein pores that permeabilize the lipid bilayer. This leads to the release of proapoptotic factors, such as cytochrome c, apoptosis-inducing factor (AIF), the secondary mitochondria-derived activator of caspases/the direct inhibitor of apoptosis-binding protein with low pI (Smac/DIABLO), HtrA serine peptidase 2 (HtrA 2), and endonuclease G (Endo G), from the mitochondrial intermembrane space to cytosol [42,44,46,47]. The prosurvival representatives of the Bcl-2 family, including Bcl-2, B-cell lymphoma-extra large (Bcl-xL), and myeloid cell leukemia-1 (Mcl-1), may prevent this scenario by binding Bax and Bak and directly inhibiting their activity [23,28,48,49,50,51,52,53]. The third subgroup of the Bcl-2 family includes the molecules containing only one BH domain (BH3 domain) and are thus called BH3-only proteins. They display proapoptotic properties, antagonizing Bcl-2, Bcl-xL, and Mcl-1, and directly or indirectly activating Bax and Bak. The BH3-only subgroup comprises the following proteins: Bim, Bad, Bik, Puma, Noxa, Bmf, Beclin-1, Hrk/DP5, and Bid [44,54,55,56,57]. The last factor is especially important, as it plays a unique role at the intersection of the two molecular cascades: the extrinsic and the intrinsic apoptosis pathways. Bid links them into the extensive molecular network, integrating the death receptor signaling with mitochondria-associated apoptotic events [45,58,59].

Bid undergoes proteolytic activation following the stimulation of the cell surface receptors [60], such as tumor necrosis factor (TNF) receptor 1 (TNFR1) [61,62], Fas [63], and TNF-related apoptosis-inducing ligand (TRAIL) receptors (TRAILRs) [30,64]. After binding the specific ligand, the receptor recruits cytosolic proteins to form a death-inducing signaling complex (DISC). This structure includes procaspase-8, the precursor of the apoptotic enzyme. Within the DISC, the zymogen undergoes dimerization and autoproteolytic cleavage, followed by the arrangement of the functional form of catalytic protein–caspase-8. This enzyme activates effector caspases (-3 and -7) as part of the extrinsic (receptor-mediated) apoptosis pathway. However, caspase-8 can also cleave Bid into p7 and p15 (tBid) fragments and, thus, affect the mitochondria downstream of the Fas, TNFR1, and TRAIL stimulation [60,65,66,67]. The larger product of the Bid proteolysis, tBid, remains integrated with the smaller one, p7, in the complex named cleaved Bid (cBid). cBid is stabilized by hydrophobic interaction between αH1 and αH3, and this non-covalent effect is strong enough to prevent the dissociation of the subunits [37,67,68,69]. The p7-tBid complex translocates to the MOM and gathers at the surface of the mitochondrion. The p7 subunit dissociates from tBid. After anchoring in the membrane, tBid undergoes conformational changes, enabling it to interact with Bax and promote the subsequent permeabilization of the MOM. Research showed that apart from Bax, tBid can activate another proapoptotic Bcl-2 family member, Bak [29,69,70,71]. In addition to the direct stimulation of Bax and Bak, tBid may also cause the indirect activation of these two proteins by competing with them for binding to the antiapoptotic representatives of the Bcl-2 family. Moreover, tBid can directly affect MOM integrity without involving Bax and Bak. Flores-Romero et al. [67] revealed that the tBid-mediated cytochrome c release might be independent of these two proteins [67].

The sequence of molecular events following MOM permeabilization involves the formation of a heptameric complex called the apoptosome. This multi-subunit structure consists of seven apoptotic protease-activating factor 1 (Apaf-1) molecules, each bound to cytochrome c. The apoptosome recruits caspase-9, which undergoes autoproteolytic activation and subsequently activates effector caspase zymogens, including procaspase-3. The active form of caspase-3 proteolyses several substrates, including the inhibitor of caspase-activated DNase (ICAD), leading to the indirect activation of CAD, DNA fragmentation, and the characteristic chromatin condensation seen in apoptosis. Another consequence of caspase-3 activation is membrane blebbing, another hallmark of type I PCD [72,73,74].

Research has revealed that Bid can also be cleaved by another proteolytic enzyme, caspase-2, in the endoplasmic reticulum (ER)-dependent pathway. Abnormal conditions, such as viral infections, can lead to an accumulation of unfolded or misfolded proteins in the ER lumen. These circumstances trigger the unfolded protein response (UPR), activate the inositol-requiring enzyme type 1 (IRE1), and stimulate the IRE1-dependent signaling cascade. The subsequent chain of molecular events may lead to the cleavage of caspase-2, which in turn catalyzes Bid proteolysis [75,76,77].

### 2.3. Targeting Bid to Mitochondria

Targeting the cBid/tBid to the mitochondria requires a series of specific molecular events. As mentioned before, an early report by Zha et al. [37] suggested the role of N-myristoylation in the conformational adaptation of the p7-tBid complex to the association with the MOM. Moreover, the team’s findings linked this post-translational modification to tBid-dependent apoptotic effects, including the release of cytochrome c from mitochondria [37]. Further reports enriched the explanation of Bid trafficking with the findings on the receptor called mitochondrial carrier homolog 2/Met-induced mitochondrial protein (MTCH2/MIMP). This structure is anchored in the MOM and exposed at the surface of the mitochondria. Zaltsman et al. [78] revealed that MTCH2/MIMP can interact with tBid, promote its recruitment to MOM, and enhance the consequent molecular events leading to the induction of the intrinsic apoptosis pathway. The team utilized mouse embryonic stem cells (ESCs) with the double knockout of the MTCH2/MIMP-encoding gene (MTCH2/MIMP-/-) to examine whether the lack of this receptor protein modifies the effect of tBid on the mitochondria. The MTCH2/MIMP-deficient ESCs displayed increased resistance to low concentrations of recombinant tBid (1nM). Meanwhile, the transfection of the double knockout mutants with the plasmid vector encoding MTCH2/MIMP substantially sensitized the cells to tBid effects. This included mitochondrial depolarization, Bak dimerization, and cytochrome c release. The team also used MTCH2/MIMP-deficient mouse embryogenic fibroblasts (MEFs) to determine the influence of the abovementioned receptor on the level of tBid-dependent apoptosis. Following the transfection with tBid-encoding adenoviral vector, the MEFs lacking MTCH2/MIMP displayed a decreased percentage of apoptotic cells, compared to MTCH2/MIMP-producing transfectants. In vitro experiments were consistent with in vivo tests performed on mouse mutants unable to produce MTCH2/MIMP in the liver. The organ-specific knockout of the *MTCH2/MIMP* gene resulted in decreased susceptibility to anti-Fas antibody treatment. Intraperitoneal injection with the abovementioned immunoglobulin caused meaningfully lowered mortality, with reduced levels of hepatocellular apoptosis, compared to *MTCH2/MIMP* gene-expressing animals. Thus, these findings suggested the hepatoprotective effect of the knockout. The subsequent molecular analyses revealed that in the context of the in vivo Fas stimulation, the lack of the MTCH2/MIMP protein impedes the tBid-MOM binding, counteracts the induction of the apoptosis execution pathway, and eventually results in lower activation of caspase-3 [78,79,80].

It is possible that the composition of the MOM bilayer, in addition to the mitochondrial surface proteins, also determines Bid’s intracellular trafficking. Several studies were devoted to the role of cardiolipin (CL) in the Bid translocation and binding to mitochondria. Lutter et al. [81] used liposomes to test how the lipid composition determines the affinity of tBid to the phospholipid membrane. They revealed that the most efficient attachment of tBid occurred in the presence of 20% CL in the target bilayer. The team also performed the in vivo experiment using the fluorescent microscopy technique and the temperature-sensitive Chinese hamster ovary (CHO) cell line transfected with the mitochondria-targeting fragment of Bid (103–162 aa) fused with the green fluorescent protein (GFP). The incubation at 40 °C significantly impaired the activity of phosphatidyl glycerophosphate synthase (PGS) within CHO cells, subsequently decreasing the CL production (compared to a control with transfectants not subjected to temperature treatment). The consequent change in the phospholipid composition of the MOM (substantial CL deficiency) thwarted the attachment of Bid (103–162 aa)-GFP fusion protein to mitochondria [81]. However, several other studies contradict the role of CL in the mechanism by which Bid targets the MOM. Analyses of the MOM lipid composition question the CL impact on Bid binding. Whereas the normal level of CL in the mitochondrial inner membrane (MIM) reaches the values of approximately 20% of the whole phospholipid content, the MOM comprises only a low amount of CL (e.g., 1.4% and 0.3% in organelles originated from rat liver and the yeast *Pichia pastoris*, respectively) [82,83,84,85]. Moreover, Choi et al. [86] showed that CL deficiency does not influence the affinity of tBid for MOM. They used RNA interference (RNAi) techniques to silence the expression of the gene encoding human CL synthase (hCLS) in the HeLa cell line. Since hCLS is an enzyme responsible for CL production, the intracellular level of this phospholipid significantly lowered after the transfection. The loss of CL rendered the cells more sensitive to apoptosis stimulation by factors such as anti-Fas antibody, staurosporine, and TNF-α combined with cycloheximide. Additionally, the confocal microscopy revealed that CL deficiency did not affect the co-localization between the exogenous tBid and mitochondria [86]. Ott et al. [82] posed the hypothesis that in artificial systems, such as liposomal ones, CL only takes over the function of some unidentified mitochondrial structures interacting with tBid in natural intracellular environments [82]. The discussion on the impact of the MOM phospholipid composition on the tBid translocation and binding remains open and further investigations are needed [79].

### 2.4. The Mechanisms of the tBid-Mediated Induction of MOM Permeabilization

As the apoptotic protein, tBid affects the permeability of the MOM, contributing to the induction of the mitochondria-dependent apoptosis pathway. The proper functionality of tBid requires the spatial rearrangement of the molecule. Using liposomes and purified mitochondria, Shamas-Din et al. [69] performed spectrofluorometric analyses of the conformational changes within the cBid/tBid molecule to trace the molecular process of membrane-bound tBid activation. In light of the findings obtained, the essential step in this sequence of events is the disintegration of cBid. The dissociation of p7 from the p7-p15 complex is followed by the spatial rearrangements of the free tBid molecule, which results in the anchoring of the latter in the MOM. The hydrophobic hairpin structure formed by αH6 and αH7 integrates into the MOM, penetrating deeply its hydrophobic interior. In addition to αH6 and αH7, two other helices, αH4 and αH5, insert themselves shallower into the phospholipid bilayer. Research showed that MTCH2/MIMP plays an essential role in tBid activation. This mitochondrial surface receptor interacts with tBid, supporting its conformational changes. The team identified several MTCH2/MIMP-binding fragments within tBid: the first one situated proximally to the N-terminus of the polypeptide, the next ones located in the neighborhood of the loop separating αH4 and αH5, and the last segment present within the C-terminal helix, αH8. The tBid-MTCH2/MIMP interplay allows tBid to gain the active, membrane-anchored spatial form that can effectively interact with another apoptotic member of the Bcl-2 protein family, Bax [69].

The latter protein is the effector molecule responsible for disrupting mitochondrial integrity. As previously mentioned, Bax plays a pivotal role in the intrinsic molecular pathway leading to apoptosis induction. Oligomeric channels formed by Bax alone or in combination with Bak permeabilize the MOM, allowing the release of cytochrome c and other mitochondrial apoptotic factors into the cytosol [30,47,87,88].

Rose et al. [89] utilized a mitochondria-mimicking planar-supported lipid bilayer to examine the interaction between tBid and Bax. Single particles were visualized confocally using an individual-particle detection and classification method. The tBid protein was found to exist in at least two conformational forms: a mobile form and a stationary one, with the former quantitatively dominating the latter. Similarly, in the presence of tBid, Bax also exhibited diverse mobility within the artificial membrane. However, the team observed that in the absence of tBid, Bax does not attach to the lipid bilayer, which suggests the crucial function of tBid in anchoring Bax to the MOM. Bax, supported by tBid, was able to form oligomers within the membrane, with oligomerization efficiency negatively correlated with protein mobility. Due to more extensive integration into the phospholipid bilayer, the stationary form of Bax exhibited an increased tendency to assemble into complexes, while the mobile form did so with reduced efficiency. Based on the findings from this study, Bax, in the presence of tBid, may organize itself into tetramers [89], which are structures of sufficient complexity to permeabilize the lipid membrane [90].

An early report by Wei et al. [62] presents the interaction of tBid with another proapoptotic effector of the Bcl-2 protein family, Bak. In an in vitro study using mitochondria isolated from the livers of Bax-deficient mice, it was revealed that the cooperation between these two molecules leads to the release of cytochrome c from the mitochondrial intermembrane space into the external environment [62]. Unlike Bax, which exists in both cytosolic and mitochondrial forms, Bak is constitutively bound to the MOM [91]. At the mitochondrial surface, its inactive form can be stimulated by the BH3 domain of tBid, resulting in Bak oligomerization, pore formation, MOM integrity disruption, and the subsequent release of cytochrome c. According to the observations of Wei et al. [62], mitochondria derived from mice carrying a *BAK* knockout did not undergo permeability transition (PT)-independent permeabilization, even in the presence of tBid. In contrast, organelles from wild-type animals released cytochrome c through a Ca^2+^-independent mechanism, demonstrating the importance of Bak in tBid-mediated MOM integrity disruption. Next, the team used a specific antibody to block the BH3-binding groove within the Bak molecule to inhibit Bak-tBid interaction. As expected, the immunoglobulin caused a significant decrease in cytochrome c release from isolated mouse mitochondria in a concentration-dependent manner. Further experiments revealed that tBid-Bak interactions were followed by a spatial rearrangement of Bak, resulting in a modified fragmentation pattern of Bak after treatment with the proteolytic enzyme trypsin. Conformational changes induced Bak to form oligomeric complexes, including dimers, trimers, and tetramers, which were detected by Western blot after stabilization with bismaleimidohexane (BMH), a crosslinker that covalently conjugates protein subunits via sulfhydryl groups. These findings were consistent with in vitro examinations. The exposure of wild-type, Bid-producing mice to anti-Fas antibody, an external trigger of the Bid-activating molecular pathway, resulted in effects similar to those observed in vitro—namely, Bak molecule conformational rearrangements and oligomerization into complexes made of three or four subunits. Meanwhile, in Bid-deficient mutant animals, Bak remained unaffected by the same treatment [62].

The direct stimulation of Bax and Bak to oligomerize in the MOM does not cover the full range of cBid/tBid apoptotic activities. cBid can support these two proteins indirectly. It competes with Bax and Bak for binding with the antiapoptotic member of the Bcl-2 family—Bcl-xL [67,92,93]. Bleicken et al. [93] utilized the fluorescence cross-correlation spectroscopy (FCCS) technique and in vitro artificial membrane systems to examine the sophisticated interplay between the three chosen molecules determining the integrity of the MOM, i.e., cBid, Bax, and Bcl-xL. The latter protein displays substantial antiapoptotic activity. In light of the findings obtained, Bcl-xL interacts with the MOM-bound Bax and disrupts the oligomerization of this protein, leading to the formation of Bax complexes with reduced size and mass. Moreover, Bcl-xL causes Bax to dissociate from the mitochondrial membrane, thus preventing the Bax-mediated MOM permeabilization. Meanwhile, cBid counteracts the abovementioned scenarios. It outweighs Bax with the affinity to Bcl-xL and traps the latter protein in cBid-Bcl-xL complexes. Bcl-xL may occur in membrane-unbound homodimeric forms. Soluble cBid can replace one of the molecules, creating an inactive cBid-Bcl-xL heterodimer. The attachment of cBid to the MOM is followed by the intensification of cBid-Bcl-xL complexing, and this process outpaces the Bcl-xL homodimerization in efficiency. The significant decrease in the pool of functional Bcl-xL reduces its impact on Bax, thus contributing to the disruption of mitochondria integrity [93]. Nuclear Magnetic Resonance (NMR) spectroscopy analyses by Yao et al. [94] provided insights into the mechanism of complexing between tBid and Bcl-xL. As observed in the studies of the team, the BH3 domain of tBid enters the hydrophobic pocket within the Bcl-xL molecule, integrating the two antagonistic proteins into a stable heterodimer [94].

Recently, research has continued to provide new insights into tBid and its role in determining mitochondrial integrity and cell viability. Flores et al. [67] demonstrated tBid’s ability to induce the permeabilization of the MOM through a mechanism independent of Bax and Bak activity. The team identified a structural adaptation in the tBid molecule that enables it to disrupt the MOM integrity without cooperation of these two proteins. Notably, helix 6 of tBid, a region with homology to the pore-forming segments of Bax and Bak, was found to be essential for the Bax/Bak-independent permeabilization of the MOM. The team used HCT AKO, a human colon cancer epithelial cell line deprived of all essential proapoptotic and antiapoptotic Bcl-2 family members, including Bcl-2, Bcl-xL, Bid, Bax, and Bak, through gene knockout. The mutants were transfected with plasmids containing *Bax*, *Bak*, or *tBid* to examine the individual impact of these proteins on cell viability. In accordance with expectations, either ectopic Bax or Bak caused apoptosis in transfected HCT AKO mutants. Surprisingly, although Bax and Bak were absent, the transfectants producing tBid also underwent apoptotic death. The ectopic tBid molecule caused the release of cytochrome c and Smac/DIABLO, followed by the signature events of apoptosis, i.e., protrusions of the plasma membrane and the condensation of chromatin into the pycnotic forms. To reveal the structural adaptation of tBid to independently permeabilize the MOM, the team examined the apoptotic properties of a tBid mutant in which two lysine residues (K157 and K158) within helix 6 were replaced by alanines. The aforementioned modifications significantly decreased the level of apoptosis in the HCK KO cells, indicating the role of helix 6 in the tBid-mediated disruption of mitochondrial integrity in the absence of Bax and Bak. Interestingly, both the confocal and stimulated emission depletion (STED) microscopy visualization of tBid’s intracellular distribution suggested that, unlike Bax and Bak, it does not form conspicuous complexes within the MOM, indicating that the oligomerization process may be less pronounced or may not occur at all [67].

According to Flores et al. [67], the Bax/Bak-independent impact of tBid on MOM integrity and cell viability may open up new therapeutic perspectives in the fight against venetoclax-resistant acute lymphocytic leukemia (ALL) cells. Venetoclax, a selective Bcl-2 inhibitor, binds to Bcl-2 and blocks its antiapoptotic activity, inducing apoptosis in cancer cells. An experiment presented in the report shed light on the mechanisms underlying the sensitivity of ALL cells without functional Bax and Bak proteins to TRAIL stimulation. The team used Nalm6 199R, a human precursor B-lymphocyte ALL cell line lacking the active Bax and Bak proteins. Nalm6 199R cells combine resistance to several proapoptotic factors, such as venetoclax and etoposide, with susceptibility to TRAIL-induced apoptosis. The aim was to elucidate the molecular basis of Nalm6 199R sensitivity to the latter cytokine. Gene knockout was performed to remove Bid from Nalm6 199R cells, followed by exposure of the Bid-deficient cells to TRAIL. The lack of Bid rendered Nalm6 199R significantly resistant to the TRAIL-mediated induction of apoptosis, indicating the role of Bid in this process. In light of the obtained findings, the promotion of tBid-dependent MOM permeabilization and the consequent apoptotic death of venetoclax-resistant cells is worth considering as a strategy to support certain cases of leukemia treatment [67].

## 3. The Role of Bid Protein in Viral Infections

The successful completion and efficacy of the viral replication cycle, the progression of host colonization, and the symptoms and complications of infection depend on cell–virus interactions, including events that determine cell viability. Different viruses can exhibit both proapoptotic [19,95,96,97] and antiapoptotic activities [19,98,99], employing diverse strategies to influence the fate of infected cells and ensure effective propagation. The deregulation of the molecular network governing cell viability contributes to phenomena such as viral protein maturation [100], immunodeficiency [101], chronic inflammation [102], and viral persistence [103]. In some cases, the virus’s influence on the mechanisms regulating cell viability can promote carcinogenesis [103,104,105,106]. Understanding the molecular events that affect apoptosis in the context of viral infections is crucial from both scientific and medical perspectives.

As one of the key players in the induction of mitochondria-dependent cell death, the Bid protein plays an important role in the course and/or potential treatment of infections caused by several pathogens, such as HBV, HSV, IAV, SARS-CoV, and HIV. Table 1 includes viral proteins involved in Bid-dependent apoptosis. The influence of diverse viruses on MOM integrity and cell viability via the regulation of Bid level, activation, and apoptotic effect is presented in Figure 1. According to the illustration, Bid links the extrinsic and intrinsic apoptosis pathways. The stimulation of death receptors by their ligands, such as TNF-α, Fas ligand (FasL), and TRAIL, induces the formation of DISC and the activation of caspase-8. This enzyme activates Bid by cleaving it into two subunits: tBid and p7. They remain non-covalently bound to form cBid. Bid can also be cleaved by caspase-2, activated in response to the ER stress. cBid migrates to the mitochondria and releases the p7 subunit. MTCH2 interacts with tBid, promoting its anchoring in the MOM and facilitating its conversion into active, functional conformation. tBid stimulates Bax and/or Bak proteins to form oligomeric pores in the MOM, allowing the release of apoptogenic factors from the mitochondrial intermembrane space into the cytosol. One of them, cytochrome c, binds to Apaf-1 to form heterodimers, which then assemble into a heptameric complex known as the apoptosome. The subsequent events include the activation of caspases-9 and -3, leading to apoptosis or, in certain cases, to gasdermin E (GSDME)-dependent pyroptosis. Additionally, cBid can bind to Bcl-xL, reducing its inhibition of Bax, thereby promoting mitochondrial dysfunction. tBid may also cause Bax/Bak-independent MOM permeabilization. Various viruses, including HBV, HSV-2, SARS-CoV-2, and IAV, can impact cell viability by modulating the Bid-mediated apoptosis pathway. While HSV-2, IAV, and SARS-CoV-2 promote the Bid (cBid/tBid)-mediated apoptotic cascade, HBV may prevent it. Pathogens can trigger the activation of caspases acting upstream of Bid proteolysis. HSV-2 infection leads to ER stress and subsequent caspase-2-catalyzed Bid cleavage. Meanwhile, the activation of caspase-8 can be mediated by open reading frame 3a (ORF3a), the proapoptotic protein of SARS-CoV-2. Polymerase basic 1 frame 2 (PB1-F2), a product of IAV, may cooperate with adenine nucleotide translocase 3 (ANT3) and voltage-dependent anion channel 1 (VDAC1) to form the mitochondrial permeability transition pore (MPTP), enhancing tBid’s effect on MOM integrity. IAV H9N2 infection upregulates the intracellular level of Bid, whereas the HBV X protein (HBx) exerts the opposite effect on the abundance of this Bcl-2 family member [37,44,45,60,67,75,78,93,107,108,109,110,111,112,113].

### 3.1. The Role of Bid Protein in HBV Infection

HBV is a representative of the *Orthohepadnavirus* genus and the *Hepadnaviridae* family. The viral proteins are encoded by a partially double-stranded, circular DNA genome, which replicates via reverse transcription and the production of an RNA intermediate. The HBV virion is composed of an icosahedral nucleocapsid surrounded by a lipid envelope adorned with glycoproteins. The viral DNA contains four open reading frames (ORFs) which partially share sequence regions and encode enzymatic and structural proteins: the viral polymerase (P); HBx; two antigens (the core (HBcAg) and the secreted HBV e (HBeAg)); and three surface proteins (small (s, HBsAg-s), medium (m, HBsAg-m), and large (l, HBsAg-l)) that are later glycosylated and exposed on the virion surface [23,24]. HBV is an etiological agent of liver diseases, including both acute and chronic hepatitis [115,116]. Long-lasting (persistent) viral infection may lead to fibrosis, which may progress to cirrhosis [117,118,119,120,121,122,123,124,125,126].

HBV may also be responsible for the development of hepatocellular carcinoma (HCC) [104,127,128,129,130,131]. The antiapoptotic properties of the HBx protein, an oncogenic factor involved in HCC pathogenesis [132], include its regulation of *BID* expression. Research has shown that HBx reduces *BID* transcript levels and Bid abundance. Chen et al. [107] compared *BID* expression in two sets of liver tissue specimens, each comprising 15 samples: an HCC group and a noncancerous collection, both derived from HBV-infected, hepatitis B surface antigen-positive (HBsAg+) individuals. The first set displayed a meaningfully decreased level of Bid protein, compared to the second one. In some cases, the abundance of this Bcl-2 family member fell below the threshold of detection by the Western blot technique. Meanwhile, the noncancerous samples exhibited a relatively high abundance of Bid, suggesting that its deficiency may play a role in HCC development. These findings were mirrored by the transcript levels in specimens from both groups. An analysis using the reverse transcriptase–polymerase chain reaction (RT-PCR) revealed that noncancerous tissue samples contained significantly higher amounts of *BID* mRNA compared to HCC specimens. Experiments on the human HCC cell line, Hep G2, confirmed these findings. The transfection of the cells with a vector encoding HBx led to a decrease in the level of Bid. Taken together, the abovementioned results enriched the molecular understanding of HCC insensitivity to external stimulators of apoptosis, such as FasL and TNF-α [107]. Earlier research by Yin et al. [108] on a mouse model linked a deficiency in the intracellular Bid to resistance to Fas-mediated cell death. Animals lacking Bid that were treated with an anti-Fas antibody displayed a very high survival rate, whereas in mice carrying a functional *BID*, the same treatment resulted in terminal outcomes due to liver damage. In vitro experiments confirmed the importance of Bid in the Fas-dependent apoptotic pathway. The knockout of *BID* rendered hepatocytes and thymocytes significantly less susceptible to the anti-Fas antibody treatment, exerting mitoprotective effects, preventing cytochrome c release, and reducing effector caspase activation. Similar observations were made in fibroblasts exposed to TNF-α [108]. The HBx-dependent alteration in Bid levels disrupts the equilibrium between proapoptotic and antiapoptotic factors. This shift skews the balance in favor of the latter in HCC cells. The oncogenic transformation of hepatocytes is accompanied by a significant increase in a range of prosurvival proteins, including Bcl-xL, Mcl-1, survivin, the X-linked inhibitor of apoptosis protein (XIAP), and the cellular inhibitor of apoptosis protein 1 (cIAP-1). The loss of Bid, which contributes to resistance against molecular triggers of the extrinsic apoptosis pathway, completes the broader molecular picture of the HBV-driven antiapoptotic and oncogenic effects on hepatocytes [133].

Interestingly, apart from Bid, another member of the Bcl-2 protein family, Bcl-xL, also contributes to the survival of HBV-associated HCC cells. Our previous review [24] described the importance of Bcl-xL in the context of virus–cell interactions. In HBV-associated HCC, an increase in the intracellular level of Bcl-xL has been reported, which, together with the loss of Bid, may contribute to the antiapoptotic effect observed in these cells [24].

### 3.2. The Role of Bid Protein in HSV-2 Infection

HSV-2 is a human pathogen classified within the *Simplexvirus* genus and the *Orthoherpesviridae* family [134]. The viral genome is composed of double-stranded DNA arranged in a linear configuration [135]. Other structural components of the virion are an icosahedral capsid, a tegument layer, and a lipid envelope studded with glycoproteins [136]. The large DNA genome harbors 74 genes responsible for the production of approximately 84 protein products, each performing diverse functions in viral replication and host cell–pathogen interactions. This rich proteome enables the pathogen to undergo both lytic and latent infections, the latter characterized by the ability to persist in a dormant state for extended periods. HSV-2, alongside HSV-1, is one of the most common representatives of the *Simplexvirus* genus, both widely distributed in the human population, extensively studied, and well characterized [137,138,139]. These pathogens are transmitted through direct contact and can cause oral or genital herpes. Notably, both herpesviruses frequently remain asymptomatic or present with mild symptoms during infection [140,141,142].

However, the aforementioned pathogens can also cause serious health problems in their hosts, which are associated with the lytic replication cycle. Due to the HSV neurotropism, the viral infection may result in severe damage to the central nervous system (CNS). In the pediatric and adult populations, HSV-1 and HSV-2 are the etiological agents of focal encephalitis and meningitis, respectively. These pathogens may also target the CNS of neonates, causing severe, widespread infections associated with substantial fatality rates [143,144].

The neuropathogenicity of HSVs is primarily attributed to their cytopathic activity. The induction of neuronal death significantly complicates the course of infection, resulting in the direct loss of neurons and dysregulated inflammatory responses within the host’s nervous system. Therefore, research into the molecular mechanisms underlying the virus-host interactions and the subsequent neurotoxicity is of utmost importance [75].

Ren et al. [75] utilized primary and secondary cell lines to elucidate the mechanisms underlying neuronal death induced by HSV, with a particular focus on HSV-2. The extended analyses revealed a connection between the Bid-dependent intrinsic apoptosis pathway and proinflammatory pyroptosis/necrosis in the context of HSV lytic infection. Human neuroblastoma-derived SH-SY5Y cells, infected with HSV-1 or HSV-2, displayed molecular hallmarks of type I PCD, including the proteolytic activation of caspase-3 and the cleavage of poly(ADP-ribose) polymerase (PARP)-1 [75]. The latter, an enzyme belonging to the ADP-ribosyltransferase (ART) superfamily, catalyzes post-translational modifications of target structures by attaching polymers of ADP-ribose, an ester molecule derived from oxidized nicotinamide adenine dinucleotide (NAD+) [145]. PARP-1 plays an important role in DNA repair due to its ability to produce a poly(ADP-ribose) scaffold for recruiting genomic maintenance proteins to sites of DNA damage [146]. Severe DNA lesions and the subsequent cellular stress can lead to PARP-1 cleavage, resulting in the release of 24 kDa and 89 kDa products, an event linked to apoptosis [147,148]. Ren et al. [75] also observed that infection with HSV-2 promotes cytochrome c efflux from mitochondria in SH-SY5Y cells. However, unexpectedly, staining with annexin V and propidium iodide did not detect apoptosis, suggesting the occurrence of pyroptosis instead. Further investigations, in light of the existing literature, helped reconstruct the sequence of intracellular events involved in HSV-2-induced neuronal death and elucidated the role of Bid (tBid) in this process. Based on the findings, excessive viral replication leads to the accumulation of unfolded or misfolded HSV-2 proteins in the lumen of the ER. The abundance of pathogen products exceeds the ER’s folding capacity, overloading the organelle and causing ER stress. These conditions initiate the UPR and induce the activation of the IRE1-dependent signaling pathway [75]. IRE1α is an ER-resident kinase that spans the ER membrane [76], facing the lumen of this organelle with the N-terminal luminal domain (NLD) exposed. This part of IRE1α is responsible for detecting misfolded proteins, either directly or indirectly in cooperation with binding immunoglobulin protein (BiP). In the latter case, BiP remains bound to NLD in an inactive IRE1α-BiP complex. Upon interaction with misfolded proteins, BiP dissociates from the NLD, signaling for IRE1α to undergo oligomerization and autophosphorylation to the active state. One of the possible molecular events downstream of the activation of IRE1α is caspase-2 cleavage [77]. Notably, HSV-2 infection was found to promote the activity of this enzyme in SH-SY5Y cells [75]. Due to its ability to conditionally cleave different substrates depending on intracellular circumstances, caspase-2 is involved in various processes, including cell cycle control and regulation, the maintenance of DNA stability, tumor suppression, and the intrinsic apoptosis pathway. This multifunctional enzyme can indirectly compromise mitochondrial integrity by catalyzing Bid proteolysis [149,150]. HSV-2 infection induces caspase-2-dependent Bid cleavage, as demonstrated by a Western blot analysis, which detected tBid in infected SH-SY5Y cells. The subsequent cascade of molecular events fits into the framework of the intrinsic apoptosis pathway, involving mitochondrial permeabilization, cytochrome c release, the apoptosome-mediated cleavage of caspase-9, and the activation of the executioner enzyme, caspase-3. The latter, in turn, cleaves PARP-1 into two fragments [75], a hallmark of type I PCD [146]. However, as previously highlighted, the final outcome of this HSV-2-dependent signaling chain is not apoptosis, but pyroptosis. As demonstrated by Ren et al. [75], the switch between these two types of cell death is driven by caspase-3, which cleaves GSDME [75]. GSDME functions as a critical executor of pyroptosis [151,152,153,154], a type of cell death that, unlike apoptosis, results in the stimulation of inflammatory response [155]. In light of the existing literature, upon proteolytic activation, gasdermins oligomerize to constitute the pores in the plasma membrane, allowing the proinflamatory cytokines, named alarmins, to exit the cells and alert the immune system [156]. Ren et al. [75] utilized SH-SY5Y cells, primary cell cultures–neural progenitor cells (NPCs), mixed mouse brain cells, and human fetal organotypic brain slice cultures (hfOBSC) to reconstruct the molecular outcomes of the HSV-2-driven cascade associated with cell death. In line with previous studies, HSV-2 infection resulted in the GSDME-mediated permeabilization of the plasmalemma and the release of the alarmin cytokine, high mobility group box 1 (HMGB1). Finally, the molecules secreted from HSV-2-infected SH-SY5Y cells mobilized pluripotent stem cell (iPSC)-derived microglia to express genes encoding proinflammatory cytokines, including interleukin (IL)-6, TNF-α, and interferon (INF)γ-induced protein 10 (IP-10). In conclusion, these findings enhance the understanding of the pathogenesis of human neural system infection with HSV-2. The ultimate manifestation of viral lytic infection is neuronal cell death, accompanied by an inflammatory response—both of which are linked to the pyroptotic pathway. Bid (tBid) plays a crucial role in the molecular cascade described above, acting as a key element downstream of ER stress induction and upstream of caspase-3 activation, an event that bridges apoptosis and pyroptosis [75].

### 3.3. The Role of Bid Protein in IAV Infection

IAV is the exclusive representative of the *Alphainfluenzavirus* genus [157] and a member of *Orthomyxoviridae* family [158]. The characteristic feature of this pathogen is the segmentation of its RNA genome, an effective adaptation to evolutionary pressures and the systematic generation of diverse serotypes and genetic variants. The single-stranded, negative-sense RNA molecule, composed of eight separated segments [159,160,161], is enclosed within an icosahedral capsid [162] encircled by an envelope, a phospholipid bilayer decorated with the viral glycoproteins, hyaluronidase (HA) and neuraminidase (NA) [163]. The versions of these two pathogen’s products are the criterion to determine the subtype of the virus [164]. Apart from HA and NA, IAV RNA encodes a minimum of eight other proteins, including three subunits of heterotrimeric viral RNA-dependent RNA polymerase (RdRp), namely the polymerase acidic (PA), polymerase basic 1 (PB1), and polymerase basic 2 (PB2) subunits; nucleoprotein (NP); two nonstructural (NS) proteins (NS1 and NS2); and two matrix (M) proteins (M1 and M2) [158]. The list of IAV’s hosts includes a wide scope of species, such as birds [165], bats [166], horses [167], pigs [168], dogs [169], seals, ferrets, and minks [24]. However, the virus is not limited to animals: IAV can cause seasonal respiratory infections in humans, with a zoonotic potential for emerging new variants [23]. The course of the IAV-driven disease is usually mild; however, severe symptoms may occasionally occur. One of the more serious manifestations is acute respiratory distress syndrome (ARDS), which is associated with a case fatality of around 40%. In certain cases, IAV infection can cause neurological complications, including encephalitis and encephalopathy [24,170,171]. The overwhelming genetic diversity of IAV strains, coupled with the evolutionary plasticity of the pathogen, remains a challenge for modern science and medicine, implicating the necessity of annual vaccinations to maintain the antiviral immunity [172,173,174].

In addition to the essential protein set, individual IAV strains can orchestrate the synthesis of supplementary products through alternative gene expression mechanisms. For example, the gene encoding the PB1 protein, a structural and functional component of the viral RdRp, may undergo a frameshift during translation. This process results in the production of PB1 frame 2 (PB1-F2), a molecule with distinct functionality compared to PB1 [166,175]. Unlike the canonical translation product, PB1-F2 does not serve as an RdRp subunit. Instead, it functions as a proapoptotic factor, immunomodulator, and proinflammatory agent. By affecting the viability of infected cells and interfering with host immune mechanisms, PB1-F2 substantially contributes to IAV virulence [166]. The immunomodulatory function of PB1-F2 includes the recruitment of leukocytes to the lungs, regulation of inflammasome formation in macrophages, induction of necrosis in these phagocytes, interference with the antiviral response mediated by mitochondrial antiviral signaling (MAVS) and retinoic acid-inducible gene 1 (RIG-I) proteins, and desynchronization of the type I interferon (IFN) response by delaying its course in favor of the pathogen [166,176].

Furthermore, another viral strategy to impair host immunity is the induction of immune cell death via the intrinsic apoptosis pathway [166,176]. Research has linked the apoptotic activity of PB1-F2 to IAV’s ability to eliminate monocytes and macrophages. The virus uses PB1-F2 to weaken the adaptive arm of the immune system and reduce the likelihood of phagocytosis [109,166,177,178].

The regulation of type I PCD also plays a crucial role in the propagation of IAV. The synthesis, degradation, and accumulation of PB1-F2 at its site of action function as a molecular timing mechanism, synchronizing cell viability with the progression of intracellular infection to the pathogen’s advantage. Efficient IAV replication has been suggested to require the alignment of apoptotic signaling with the viral replication cycle. While the premature death of the infected cell could disrupt this process, caspase-mediated proteolysis contributes to the processing of viral proteins, such as NP [100,109,179]. In a study by Wurzer et al. [100], the use of Z-DEVD-FMK, a caspase-3 inhibitor, significantly reduced the viral yield in IAV-infected A549 cells [100].

Zamarin et al. [109] elucidated the role of Bid (tBid) in PB1-F2-mediated apoptosis, highlighting its potential indirect contribution to both the impairment of the host antiviral response and the facilitation of efficient viral propagation. In light of the findings, PB1-F2 migrates to the mitochondria, where it self-assembles into homo-oligomeric pores within the MOM. Additionally, it may cooperate with ANT3 and VDAC1—transmembrane structures localized within the OMM and IMM, respectively—to induce the MPTP. Moreover, PB1-F2 was found to synergize with tBid in disrupting the integrity of the MOM. Experiments with isolated mouse liver-derived mitochondria demonstrated the cooperation of these two proteins in the evacuation of cytochrome c from the intermembrane space. The recombinant tBid-mediated release of the aforementioned factor was approximately ten times higher when preceded by preincubation of the organelles with PB1-F2. The team hypothesized that the mechanism of intrinsic apoptosis pathway induction in IAV-infected cells is based on the synergistic relationship between PB1-F2 and apoptotic members of the Bcl-2 protein family, including tBid. According to this hypothesis, the role of PB1-F2 is to increase the mitochondria’s susceptibility to permeabilization by tBid, Bax, and/or Bak [109].

In addition to the proapoptotic activity of PB1-F2 and tBid in IAV-infected cells, antiapoptotic representatives of the Bcl-2 family, including Bcl-xL, may participate in the regulation of cell viability in the course of viral replication. Research suggests that Bcl-xL, together with Bcl-2 and Bcl-w, counteracts MOM permeabilization at the early stage of infection, preventing premature apoptosis and promoting viral replication [24,180]. Notably, Bcl-xL may also play a role in the later stages of infection. Then, the upregulation of this protein via the Janus kinase/signal transducers and the activators of transcription (JAK-STAT) pathway has been linked to a switch from apoptosis to pyroptosis, an inflammatory form of cell death that favors viral clearance [24]. These findings highlight the dynamic balance between pro- and antiapoptotic Bcl-2 family members in the progression of IAV infection.

The influence of PB1-F2 on mitochondria integrity appears to extend to the scale of the whole infected cell. A team sought to determine whether transfection with PB1-F2 affects the susceptibility of epithelial-like secondary cells, 293T, to diverse apoptotic stimuli. The test suggested that upon exposure to UV radiation, cisplatin, or TNF-α, the presence of this viral protein enhances the process of caspase-3-mediated PARP proteolysis, an apoptosis hallmark already mentioned. Interestingly, the immunofluorescent visualization of the cleaved form of cytokeratin, another marker of type I PCD, demonstrated that A549 transfectants overproducing Bcl-xL exhibited resistance to ectopic PB1-F2-dependent sensitization to TNF-α [109]. Thus, this observation suggested that the antisurvival signaling induced by the aforementioned cytokine in PB1-F2-positive cells involved apoptotic members of the Bcl-2 family: the Bid protein functions as the common link of the two apoptosis pathways, extrinsic and intrinsic ones, and Bax and/or Bak function as the MOM-permeabilizing agents. Meanwhile, Bcl-xL, an antagonist of tBid, Bax, and Bak, counteracts their apoptotic function, mitigating the cell’s sensitization to TNF-α, as observed in the experiment [109,181,182,183]. To sum up, Bid (tBid) appears to promote viral infection by synergizing with PB1-F2 in triggering cell death in IAV-infected cells. This PB1-F2/tBid-dependent apoptosis facilitates viral propagation and impairs the host immune response, contributing to leukocyte depletion and disrupting TNF-α antiviral signaling [109].

Bid (tBid) has also been implicated in developing gut infection symptoms during infection with IAV H9N2. This virus subtype primarily infects birds but can undergo zoonotic transmission to humans, leading to gastrointestinal manifestations. In a study by Qu [110], IAV H9N2-infected human colorectal adenocarcinoma HT-29 cells, used as an in vitro model for IAV infection, exhibited elevated levels of tBid, as well as the increased activation of caspases-8 and -9, enzymes acting upstream and downstream of Bid cleavage. These findings confirmed the role of Bid in linking the extrinsic and intrinsic apoptosis pathways in the context of IAV infection. Further molecular analyses revealed the significant influence of IAV H9N2 infection on cytokine production in HT-29 cells. Specifically, the team observed an increase in the levels of IL-1 and IL-8, as well as a marked upregulation of C-C motif chemokine ligand 5 (CCL5) and IP-10. The high levels of proinflammatory cytokines suggested that IAV H9N2 intestinal infection induces a significant host immune response. In vitro observations were supplemented with experiments using a murine model. The systemic IAV H9N2 infection affected not only the respiratory but also the gastrointestinal system, causing a wide range of symptoms, including—in a subset of animals—diarrhea and severe edema in the intestines. These observations imply that, in certain cases, IAV H9N2 may disrupt the integrity of the intestinal epithelium, potentially allowing the gut to serve as a route of infection. Taken together, these findings suggest that tBid-mediated, mitochondria-dependent apoptosis may contribute to gastrointestinal distress in IAV H9N2-infected hosts by eliminating epithelial cells—thus impairing the intestinal barrier [110].

### 3.4. The Role of Bid Protein in SARS-CoV-2 Infection

SARS-CoV-2 is a member of the *Betacoronavirus* genus and the *Coronaviridae* family [184]. Its proteins are encoded by a single-stranded, positive-sense RNA molecule. The SARS-CoV particle consists of a helically symmetric nucleocapsid and a lipid envelope [23,24] adorned with the highly immunogenic trimeric spike glycoprotein (S), which gives the virion a crown-like shape [185]. The viral genome includes 14 open reading frames (ORFs) that encode 29 products, including 4 structural proteins: S (mentioned above), envelope (E), membrane (M), and nucleocapsid (N); 16 nonstructural proteins, such as viral polymerase (RdRp); and 9 accessory proteins, such as ORF3a [186]. SARS-CoV-2 is well known as the causative agent of COVID-19, the outbreak of which was declared a global pandemic by the World Health Organization (WHO) in March 2020 [23]. The virus causes a respiratory tract infection that may progress to a systemic scale and manifest with symptoms in various locations [187]. The course of the human host–pathogen interaction can be mild; however, it can also lead to serious, life-threatening manifestations, including ARDS and multiple organ failure [188].

SARS-CoV-2 may exert both prosurvival and proapoptotic effects, with its impact on host cell survival being context-dependent. The strict regulation of cell viability during infection facilitates the successful propagation of the infectious agent. By affecting miscellaneous molecular pathways, the pathogen harmonizes apoptotic signaling with its strategy of viral replication and host organism colonization. SARS-CoV-2 infection interacts with the Bcl-2 family protein signaling network, either counteracting or promoting the intrinsic apoptosis pathway [23,111,114,189,190]. An example of the virus’s prosurvival effect on these molecules is the upregulation of the intracellular level of the mitoprotective factor Mcl-1. Specifically, the viral N protein facilitates the deubiquitination of Mcl-1 at lysine 63 (Lys63), enhancing the stability of this antiapoptotic protein. This, in turn, preserves mitochondrial integrity and prolongs the viability of infected cells. Consequently, SARS-CoV-2 propagation is supported, as evidenced by significantly higher viral titers in the human colorectal adenocarcinoma cell line expressing the N protein (Caco2-N) compared with cells treated with an Mcl-1 inhibitor [23,189]. Moreover, the antiapoptotic activity of SARS-CoV-2 contributes to the frequent asymptomatic nature of the infection. By delaying or suppressing apoptosis, the virus impedes the early recognition of the disease and its etiological agent, thereby facilitating its spread within the host population [23].

Conversely, SARS-CoV-2 can also promote type I PCD under certain conditions, demonstrating the pathogen’s ability to modulate apoptosis-related signaling cascades in distinct ways, depending on the site and stage of infection or the cellular context [114]. Meanwhile, research into a transgenic mouse model has shown that SARS-CoV-2-driven mitochondria-dependent cell death may negatively affect the host’s lung condition, contributing to the development of pulmonary injury, hemorrhage, and inflammation within lung tissues [111]. Interestingly, the attenuation of apoptosis through the proapoptotic nucleolin (NCL) inhibitor diminishes the efficiency of the pathogen’s propagation, as evidenced by the lower viral titer in hamster pneumocytes [191]. In addition to Mcl-1, another Bcl-2 family member important for the pathogenesis of SARS is Bcl-xL. Research suggests that the E protein of SARS-CoV-2 may bind to Bcl-xL, reducing host cell viability and potentially contributing to virus-associated lymphopenia [24]. Thus, both the anti- and proapoptotic activities of SARS-CoV-2, exhibited depending on the circumstances, may contribute to viral reproducibility and pathogenicity [23,189,191].

The antisurvival viral products include ORF3a accessory protein [112,113,114]. Although initially considered a viroporin capable of permeabilizing the plasma membrane by forming ion-conducting channels [114], recent cryo-EM studies have challenged this model, suggesting that ORF3a does not function as a classic viroporin but instead affects endolysosome formation, cellular trafficking, and immune evasion mechanisms [192]. ORF3a is known to induce ER stress-dependent apoptosis, which may facilitate viral propagation [114]. Moreover, this viral product has been found to directly induce caspase-8 activation, leading to subsequent Bid cleavage. The downstream signaling cascade triggered by this molecular event is a characteristic consequence of Bid proteolysis; it includes MOM permeabilization, cytochrome c release, and the sequential proteolytic activation of caspases-9 and -3, which are associated with the intrinsic apoptosis pathway [112,113,114,193,194,195]. This scenario has diagnostic implications, as DNA material derived from disrupted mitochondria can be detected in the blood of SARS-CoV-2 infected patients and may be associated with an unfavorable COVID-19 prognosis [196,197,198]. Ren et al. [112] demonstrated ORF3a-mediated caspase-8 cleavage, subsequent Bid activation, and other apoptotic events in two kidney epithelial cell lines—Vero (derived from the African green monkey) and HEK293T (human)—as well as in HepG2 HCC cells, each transfected with a plasmid vector encoding ORF3a. The team also investigated regulatory amino acid sequences within the ORF3a molecule that govern its intracellular distribution. Mutations in either the cysteine-rich fragment (C130/C133) or the tyrosine-based sorting motif (Y160) disrupted ORF3a’s association with the plasma membrane, observed for the wild-type protein, and caused the modified viral proteins to localize in the cytosol. This altered localization significantly reduced, though did not entirely abolish, ORF3a’s ability to activate caspase-8, thereby indirectly impairing Bid cleavage and the downstream apoptotic cascade culminating in caspase-3 proteolysis. These findings underscore the critical role of the two membrane-targeting motifs in preserving ORF3a’s Bid-dependent apoptotic properties [112]. In summary, Bid serves as a link between the extrinsic, receptor-mediated apoptosis pathway and the intrinsic, mitochondria-dependent proapoptotic cascade in the context of SARS-CoV-2 infection. The antisurvival activity of Bid (tBid) may contribute to the virus’s pathogenicity and replicative capacity, as type I cell death can account for certain COVID-19 symptoms and, in an appropriate context, facilitate the pathogen’s propagation [111,112,191].

## 4. Therapeutic Potential of Bid (tBid) Protein in Apoptosis Regulation: Implications for Viral Infections and Virus-Associated Diseases

A deep insight into the intracellular molecular network regulating apoptosis, with a special emphasis on the role of the Bcl-2 protein family, is crucial in addressing contemporary medical challenges. As mentioned before, the vitality of the infected cell may shape the dynamics of viral replication and, on a broader scale, influence the progression of systemic infection, its health consequences, and the overall fate of the host. The Bid protein, particularly in its functional, activated form (tBid), is considered a potential weapon against intracellular pathogens. Due to its proapoptotic activity, this factor can influence cell viability while simultaneously disrupting viral replication, which is dependent on the resources of the host [199,200,201,202].

### 4.1. The Therapeutic Potential of tBid in HBV-Associated HCC

One of the most devastating and life-threatening complications of chronic HBV infection is the development of HCC [203,204,205,206,207,208,209,210,211,212]. The treatment of HCC can be either curative or palliative, depending on the stage of tumor development and prognosis. As HCC is resistant to chemotherapy, medical interventions include invasive surgical methods, such as partial or major hepatic resection and liver transplantation. These approaches offer hope for survival at the early stages of tumor development; however, HCC is often diagnosed at more advanced stages, in which treatment options are more limited. Furthermore, liver transplantation presents the challenge of finding a suitable donor, while partial hepatic resection carries the risk of recurrence and peri- or postoperative mortality due to an insufficient mass of the remaining liver for proper regeneration [213,214,215].

Alternative curative methods for HCC therapy include chemical and physical ablations, such as ethanol or radiofrequency ablation. These approaches have the advantage of reducing systemic side effects; however, a significant limitation is the higher recurrence rate compared to resection. The condition of the liver, the stage at diagnosis, the size of the tumor, the number of foci, and the metastatic potential of the neoplasm are key factors that determine whether a patient is classified for curative or palliative treatment. Due to poor prognoses, the majority of individuals with HCC fall into the latter category. They may undergo treatment through embolization, systemic chemotherapy, and/or molecular targeting. The first method is based on obstructing the blood supply to the tumor tissue and may be supported by cytostatic drugs, such as doxorubicin and cisplatin, or combined with radiotherapy. However, systemic chemotherapy and radiation therapy show relatively low efficacy in treating HCC and are associated with a wide range of side effects. Molecular targeting using antitumorigenic and angiogenic agents, such as sunitinib, linifanib, brivanib, and sorafenib, faces several challenges, including limited efficacy, low effectiveness against metastatic tumors, high treatment costs, liver toxicity, and numerous side effects [213,214,215].

As highlighted earlier, the Bid protein, naturally synthesized in hepatocytes, may be targeted by the viral protein HBx, and a significant deficiency of this proapoptotic member of the Bcl-2 family is characteristic of HCC phenotypes [107]. The aforementioned observation, together with the overall knowledge about the Bid’s antisurvival activity, supports the idea of using it in anticancer therapy. Miao et al. [199] performed an empirical evaluation of the effectiveness of tBid in the fight against HBV-associated HCC. In vitro experiments revealed that the ectopic synthesis of tBid leads to the apoptosis of HCC cells. Transfection with an adenoviral vector encoding this protein effectively killed transformed hepatocytes representing two human cell lines, the p53-resistant Hep3B and the p53-sensitive PLC/PRF/5. Exogenous tBid promoted apoptotic events in transfectants, including cytochrome c release and the activation of caspases-8, -9, and -3. Next, the team expanded their research to include the in vivo examination of a tBid-encoding vector in combating HCC. Mice were implanted with Hep3B cells and transfected with the vector, and the ectopic expression of tBid induced apoptosis in transformed cells, inhibiting tumor development [199].

A critical challenge in cancer therapies is achieving proper selectivity. The strategy to target cancerous cells without affecting healthy ones may involve distinguishing the former through specific molecular markers. HCC cells, unlike non-transformed hepatocytes, tend to produce α-fetoprotein (AFP), a fetal glycoprotein that is synthesized, among other sites, in the liver during prenatal and neonatal development, but later decreases in level and is typically produced in minimal amounts in adults [216]. Therefore, the team of Miao et al. [199] used the adenoviral vector with an AFP-induced promoter, conditioning the synthesis of tBid upon the presence of the aforementioned HCC marker. Indeed, in in vitro conditions, the treatment selectively affected AFP-positive HCC cells (Hep3B and PLC/PRF/5), leaving the AFP-negative ones (Chang liver cells) intact. Meanwhile, in an in vivo examination using the aforementioned mouse model, the therapy targeted the tumor without causing observable injury to the neighboring liver tissue or to important distant structures, including the kidneys, heart, and lungs, thus demonstrating its selectivity. The findings of the team provided the basis for considering ectopic tBid as a potential weapon in the fight against HCC, either alone or in conjunction with chemotherapeutic agents (such as doxorubicin, TNF-α, 5-fluorouracil) and/or radiotherapeutic interventions [199].

The concept of using the tBid protein in HCC therapy was further developed by Yan et al. [200]. The team constructed a hybrid protein composed of four subunits, each playing a critical role in the therapeutic strategy. The molecule contained a single-chain variable fragment antibody (scFv15), a furin-cleavable diphtheria toxin motif (Fdt), hemagglutinin subunit 2 (HA2), and tBid. The first element of this complex structure ensures the selectivity of its action due to its specificity for the viral glycoprotein s (HBsAg-s) [200]. As mentioned previously, HBsAg-s is present in the lipid envelope of the HBV virion [217], and it may also be exposed on the surface of infected cells, where it can be recognized by scFv15. This characteristic directs the therapy specifically to HBV-positive hepatocytes. In addition to HBsAg-s, another molecular marker, furin, was used to differentiate HCC cells from healthy ones [200]. This enzyme is overproduced in cancer cells [Huang], and thus, the next subunit of the scFv15-Fdt-HA2-tBid hybrid protein is designed to trigger the cytotoxic effect in response to the abundance of this molecular marker. The furin-catalyzed cleavage of the scFv15-Fdt-HA2-tBid molecule at the Fdt region is essential for activating the hybrid, thereby limiting its action to cancer cells that overproduce the enzyme. HA2, the third component of the therapeutic protein, enables it to escape from the endosome after internalization. Finally, the last subunit, tBid, induces apoptosis in transformed infected cells. The protein designed in this way turned out to display high effectiveness in the battle against HCC, both in vitro and in vivo. The team determined the contribution of HA2 to the hybrid’s efficacy by comparing the cytotoxic effects of the complete molecule with its simplified HA2-lacking version (scFv15-Fdt-tBid) on three HCC cell lines: HepG2, Hep3B, and PLC/PRF/5. The comparative analysis was designed to address whether HA2 interferes with the tBid apoptotic activity or, conversely, synergizes with it (and the rest of the hybrid’s fragments) to maximize the therapeutic effect of the drug. The cytotoxicity of both proteins depended on the tested cells’ ability to synthesize HBsAg-s, demonstrating the selectivity of the construct. The apoptotic effect was highest in HBsAg-overproducing PLC/PRF/5 hepatocytes and limited in Hep3B cells, which have a lower level of viral glycoprotein, while in antigen-lacking HepG2 cells, both hybrids remained inefficient. Both the complete molecule and the three-subunit alternative exhibited significant efficacy in combating PLC/PRF/5 cells. However, the presence of HA2 in the full hybrid reduced the half-maximal inhibitory concentration (IC50) by approximately 10-fold compared to the IC50 of the simplified HA2-lacking version (scFv15-Fdt-tBid). Thus, the in vitro examination highlighted the advantages of the complex anti-HCC drug candidate: high precision, optimal IC50, partly due to HA2-dependent evacuation from endosomal vesicles, and notably, the cytotoxicity attributed to the apoptotic properties of tBid [200].

Importantly, the synergy between the parts of scFv15-Fdt-HA2-tBid was also reflected in the mouse model. PLC/PRF/5 cells carrying the luciferase gene were implanted into animals, and subsequent tumor growth was monitored under treatment with scFv15-Fdt-HA2-tBid or scFv15-Fdt-tBid molecules. In vivo bioluminescence imaging (BLI) revealed that the full protein inhibits the expansion of the cancerous mass with significantly higher efficiency (95%) compared to the HA2-lacking hybrid (75%) [200].

### 4.2. The Therapeutic Potential of tBid in HIV Infection

HIV is a representative of the *Ortohepadnavirus* genus and the *Hepadnaviridae* family [23]. The viral proteins are encoded by a single-stranded, positive-sense RNA genome [218], which replicates through reverse transcription and the synthesis of a double-stranded DNA copy (cDNA) intermediate [219]. The latter integrates into the host genome [220]. The HIV virion is composed of the two identical RNA molecules and a cone-shaped capsid [221], which is surrounded by a lipid envelope [23]. Trimeric glycoproteins (gp), gp120 and gp41, are exposed on the viral surface [222]. Each RNA copy encodes a minimum of 15 proteins, among which are reverse transcriptase (RT), gp120/41 precursor, gp160, and other enzymatic, structural, or accessory products [23,222]. The latter category includes the viral protein R (VpR) [223], the trans-activator of transcription (Tat) [224], and the regulator of the expression of virion proteins (Rev) [225]. HIV is well known as the causative agent of acquired immunodeficiency syndrome (AIDS), a severe disease that, while treatable, remains incurable. If left untreated, AIDS has a terminal outcome [226,227].

The gp120/41 complex binds to the cluster of differentiation (CD)4 receptor, enabling the virus to target CD4+ cells, such as T helper (Th) lymphocytes and monocytes/macrophages [222]. These cells facilitate HIV’s systemic dissemination and serve as viral reservoirs, within which the virus evades the host’s antiviral response and anti-HIV treatments. The long-term evasion of elimination is possible because, after infecting immune cells, HIV can establish and maintain a state of latency. This capacity not only allows HIV to spread systemically and survive despite the antiviral response, but also has significant clinical and epidemiological consequences, posing challenges to modern science, medicine, and disease control policies. A characteristic feature of AIDS is its prolonged asymptomatic or mild course, which can obscure the actual cause of infection and hinder an accurate diagnosis. The challenges of the early detection of HIV infection favor its transmission within populations. Therefore, gaining a deeper understanding of the molecular mechanisms underlying the establishment of latency is of the highest priority [23].

Meanwhile, it has been shown that HIV can infect monocytes/macrophages without significantly affecting their viability [23,228,229]. In vitro studies revealed that during the maturation of HIV-infected monocytes into macrophages, the viral envelope glycoproteins (gp120/gp41) upregulate the production of macrophage colony-stimulating factor (M-CSF), a cytokine that is involved in the increase of intracellular levels of Bcl-2 family prosurvival members, including Mcl-1, Bcl-xL, and Bcl2-related protein A1 (Bcl2-A1). This upregulation results in the significant desensitization of infected cells to external apoptotic stimuli, such as TRAIL. In light of the obtained results, HIV appears to indirectly modulate the apoptosis molecular network by suppressing its mitochondria-dependent branch, which helps maintain the viral reservoir [23,230,231].

A deep insight into the HIV–host interaction provides the foundation for developing antiviral therapeutic strategies aimed at promoting the intrinsic proapoptotic cascade. In vitro studies by Huelsmann et al. [201] focused on the preliminary evaluation of the therapeutic potential of a tBid-encoded DNA vector, pLRed(INS)2R (pLtBid(INS)2R), specifically designed to deliver the proapoptotic Bcl-2 family member to HIV-1-infected cells. The team addressed key challenges in the development of effective treatments for viral infections, including the potential immunogenicity of antiviral agents and the critical need for their selective action. tBid, as a natural component of the cellular proteome, does not induce an immune response, demonstrating a significant advantage over xenogeneic antiviral factors. Meanwhile, specificity was achieved by placing the tBid-encoding gene under the control of an inducible promoter. The selective treatment is ensured by the regulation of expression through the 5′ and 3′ long terminal repeats (LTRs), the Rev-responsive element (RRE), and the inhibitory sequences (INS) from the *gag* gene. All of these sequences were borrowed from HIV. Thus, the presence of Tat, a viral protein essential for HIV-1 DNA transcription, was a prerequisite for ectopic tBid synthesis in donor cells transfected with the pLtBid(INS)2R vector. Moreover, the inheritance of RRE and INS required the availability of Rev, a viral protein responsible for the efficient transport of the transcript from the nucleus to the cytosol, where translation occurs. The linking of tBid-encoding gene expression, both at the transcriptional and translational levels, to the presence of functional viral products enabled the distinction between HIV-1-infected and HIV-1-negative cells. In vitro experiments on human HeLa SS6 cells confirmed the specificity of pLtBid(INS)2R expression for transfectants producing both Tat and Rev, as cells lacking either of these proteins were subject to negative selection. The cotransfection of donor cells with pLtBid(INS)2R and the DNA vector encoding the full set of viral products, along with green fluorescent protein (GFP)—specifically, the HIV-1 proviral clone pNL4-3/GFP—fulfilled all the necessary criteria for ectopic tBid synthesis. Indeed, double transfectants underwent immediate apoptosis, before progeny virions could assemble or be released. To sum up, pLtBid(INS)2R has proven to be a promising, highly selective solution for combating HIV-1 infection, and further research into its potential utility in gene therapy is highly recommended [201].

As previously discussed, current medical approaches have encountered limitations in the fight against HIV, making complete eradication from the host organism unattainable. The symptomatic treatment of HIV patients, while significantly reducing mortality, does not offer a cure for AIDS. The reason for this situation lies in HIV latency within CD4+ immune cells, where the virus persists as an unexpressed proviral DNA form integrated into the host genome. Current anti-HIV treatment strategies are based on antiretroviral therapy (ART), which includes nucleotide reverse transcriptase inhibitors (NRTIs), protease inhibitors (PIs), integrase strand transfer inhibitors (INSTIs), entry inhibitors (EIs), and C-C chemokine receptor type 5 (CCR5) antagonists. The advantages of ART include the significant elongation of the patient’s lifespan, reduction in viremia, strengthening of the host immune system, and an increase in the number of CD4+ lymphocytes, which diminishes the risk of AIDS development. However, as mentioned earlier, the pathogen is not fully eliminated and remains hidden in cells. Additional limitations and disadvantages of ART include side effects such as gastrointestinal issues, liver and/or kidney failure, hypercholesterolemia, and an elevated susceptibility to cardiovascular diseases [232,233,234,235,236]. The principle behind certain anti-HIV therapies is to stimulate the virus to transition into a productive infection, effectively “bringing it out of hiding”. This strategy increases viral activity, but its objective is to reduce the latent virus reservoir by facilitating the clearance of reactivated virus, for example, through the immune system or ART, potentially achieving a functional cure. For instance, Klinnert et al. [202] have evaluated the integration of tBid-based antiviral treatment within the complex shock-and-kill antiviral strategy. The proposed scheme of action included reactivating the pathogen from latency, thereby enabling the opportunity to target it selectively. The clustered regularly interspaced short palindromic repeats (CRISPR)/dead CRISPR-associated 9 (dCas9)-VpR activation (CRISPRa) system, guided by RNA V (gRNA V), was ectopically expressed in J-Lat 10.6 cells—childhood T-cell acute lymphoblastic leukemia (T-ALL) cells harboring latent HIV-1 infection—to recognize the provirus and stimulate HIV-1 gene transcription. Consequently, the presence of the two previously mentioned accessory proteins, Tat and Rev, enabled the activation of the tBid-encoding gene under an inducible promoter. The Tat- and Rev-specific synthesis of tBid triggered apoptosis in transfected cells, thereby preventing further viral propagation. The treatment’s precision relied on the delivery of the CRISPRa system and tBid via a CD3-retargeted adenoviral vector, which selectively targeted CD3+ immune cells, as CD3 is a marker predominantly found on T lymphocytes [202].

Summing up, the potent, dynamic, and rapid apoptotic activity of tBid, its endogenous nature, and consequent non-immunogenic profile render it a promising weapon in combating HIV—a pathogen that bases its immune evasion strategy on latency and the maintenance of infected cell viability [201,202].

### 4.3. Challenges and Limitations of Potential tBid-Based Therapies

As mentioned above, tBid-based approaches have been primarily explored in the context of potential treatments for HBV-associated HCC and HIV infection. Current therapies face several limitations, which motivate the search for more effective solutions. Bid plays a critical role in regulating cell viability, modulating the response to stress stimuli, and maintaining homeostasis in multicellular organisms. Its active form, tBid, exhibits potent apoptotic properties, positioning it as a promising therapeutic agent for viral infections. While it offers hope for advancing antiviral therapies, it also faces significant limitations and challenges. Specifically, while promoting apoptotic signaling may facilitate viral clearance, it could also result in tissue damage, inflammatory responses, and/or compromise the host immune system. For example, the complexity of tBid-based therapies would be further amplified in the context of neurotropic viruses, where the use of tBid to induce apoptosis in neural tissues could lead to unintended consequences, such as promoting neuroinflammation and exacerbating neuronal death. These risks underscore the need for a highly selective, context-specific approach that considers the specificity of virus–cell interactions, the physiology of host tissues, and the dynamics of the molecular apoptosis regulatory network within the nervous system. The intracellular interplay between the Bid-dependent intrinsic apoptosis pathway and other molecular cascades, such as the ER stress response [75], poses a significant obstacle to the potential use of tBid-based therapies in certain viral infections. In the treatment of HSV-2, this approach could lead to GSDME-mediated pyroptosis, resulting in neuronal loss and a strong inflammatory response. Moreover, delivering tBid-based therapeutics for viral infections in neural tissues faces the challenge of the blood–brain barrier (BBB), which blocks access to the brain [237]. This necessitates the development of strategies to cross the barrier without causing detrimental side effects. Since infected cell death can either benefit or harm the host, using proapoptotic molecules like tBid in the treatment of neural infections carries the risk of far-reaching health consequences, including neurodegeneration.

Preliminary research using a mouse model to evaluate the effectiveness of tBid-based therapy for HCC treatment supports the need for further exploration. The necessity for selectivity in this approach can be addressed by incorporating an AFP-specific promoter within the tBid-encoding vector, thereby preventing side effects, as demonstrated in xenograft mice. Following tBid-based gene therapy, these mice showed a substantial decrease in tumor volume, without detrimental effects on healthy surrounding tissues or organs such as the liver, heart, lungs, and kidneys [199]. Additionally, as previously discussed, the specificity of HCC treatment could be achieved by creating a tBid hybrid with scFv15 and Fdt [200]. Similarly, in the case of HIV, in vitro studies suggest that the pLtBid(INS)2R plasmid [201], or the CD3-retargeted adenoviral vector encoding tBid, as well as the CRISPRa system, could serve as potential tools to enhance selectivity [202]. While these findings are promising, further research—particularly clinical trials—is essential to address critical factors such as the efficiency of tBid delivery, potential side effects, and the stability of the system in complex in vivo environments. Notably, the adaptation of these strategies from in vitro and animal models to human therapies presents a significant medical challenge. This translation may face obstacles such as potential immune responses, off-target effects, and concerns regarding the long-term effectiveness of the treatment. Therefore, the potential widespread implementation of these approaches must be preceded by comprehensive clinical trials in humans to evaluate both the safety profile and eventual efficacy.

## 5. Conclusions

The Bid protein is a signaling molecule of significant importance, as it integrates two apoptosis pathways: the extrinsic and intrinsic cascades. It remains inactive in the cytosol until proteolytic activation by caspase-8 or -2, triggered by death receptor engagement or ER-mediated stress, respectively. At the MOM, in its cBid/tBid form, it can induce permeabilization either indirectly or directly by stimulating Bax and/or Bak to form oligomeric pores in the lipid bilayer. The subsequent release of the apoptogenic factor cytochrome c from the mitochondrial intermembrane space to the cytosol results in apoptosome assembly and the sequential activation of caspases-9 and -3.

The equilibrium between apoptotic and prosurvival signaling allows for the regulation of cell viability in response to the needs of the multicellular organism. Abnormalities in this balance may contribute to the development of various pathological conditions. Among other disease factors, viruses can interfere with apoptotic signaling, disrupting the survival/death equilibrium in their favor. The Bid protein plays a crucial role in infections caused by various pathogens, including HBV, HSV, IAV, and SARS-CoV-2. Depending on the virus, site and stage of infection, or cellular context, they may either counteract or promote apoptosis, thus leading to the downregulation or upregulation of Bid, respectively, in terms of its levels/activity.

Bid plays a pivotal role as a molecular linker that integrates two apoptotic pathways: the extrinsic and intrinsic. The intersection of these signaling cascades is crucial for maintaining cellular homeostasis and responding to stress stimuli. Bid is involved in regulating the balance between cell death and survival, adjusting this equilibrium to the varying needs of the multicellular organism. However, viruses can influence cell viability in ways that favor the development of infection, promoting viral propagation, immune evasion, and pathogenesis, with consequences such as tissue damage and oncogenesis. By modulating Bid levels and activity, either synergizing with or antagonizing it, viruses enhance or disrupt the bridge between these two apoptosis pathways. This alters the holistic apoptotic signaling network in a manner that benefits the virus while harming the host.

The modulation of Bid signaling during intracellular infection contributes to the development of certain diseases and pathological states. In HBV-associated HCC, the downregulation of Bid levels mediated by the viral HBx protein desensitizes cells to extrinsic apoptotic stimuli, such as FasL and TNF-α, thereby promoting oncogenic transformation. On the contrary, in HSV-2, IAV, and SARS-CoV-2 infections, the enhancement of Bid signaling may represent a pathogenic event. Bid (tBid) has been identified as a key contributor to the neurotoxicity and proinflammatory effects of HSV-2. By compromising MOM integrity in lytically infected neurons, tBid links HSV-2-induced ER stress to caspase-3 activation. This cascade leads to GSDME cleavage, which permeabilizes the plasma membrane, releases alarmins, and shifts the outcome from apoptosis to pyroptosis. Consequently, the death of neural cells is accompanied by an inflammatory response mediated by cytokine production in microglia. As a result, Bid emerges as a central molecule in the intricate signaling network driving CNS inflammation and pyroptotic cell death, processes characteristic of the severe neural damage associated with diseases like meningitis. In IAV infection, the synergistic effect of tBid and the viral product PB1-F2 on mitochondrial permeability may promote viral propagation and weaken the host immune response, contributing to leukocyte depletion and disrupting TNF-α antiviral signaling. It has also been suggested that the IAV H9N2-driven increase in intracellular Bid levels may contribute to mitochondria-dependent apoptosis in intestinal epithelial cells, potentially impairing the intestinal barrier and promoting distress, manifested by diarrhea and severe edema in the intestines. Conversely, in the course of SARS-CoV-2 infection, the viral protein ORF3a indirectly stimulates Bid cleavage by the activation of the upstream enzyme, caspase-8. The proapoptotic activity of Bid (tBid) may enhance the pathogenicity and replication of the virus, as mitochondria-dependent type I cell death is implicated in certain COVID-19 manifestations and, under specific circumstances, may facilitate SARS-CoV-2 propagation.

A deep understanding of the intracellular apoptosis molecular network, with a focus on the Bcl-2 protein family, is pivotal in addressing modern medical challenges. The apoptotic potential of the active form of Bid, tBid, makes it a valuable tool in the fight against viruses, particularly in the treatment of chronic, persistent, and/or latent infections, where the survival of host cells may benefit the pathogen.

Preliminary studies demonstrated the effectiveness and selectivity of tBid-based therapeutic approaches in combating HBV-associated HCC and HIV infection. For HBV-related HCC, an adenoviral vector encoding tBid under the control of an AFP-induced promoter, as well as the scFv15-Fdt-HA2-tBid hybrid protein, exhibited remarkable selectivity and induced apoptosis in AFP-positive cells while sparing AFP-negative ones. In the case of HIV infections, the pLtBid(INS)2R DNA vector selectively triggered apoptosis in infected cells by linking tBid expression to the presence of viral proteins Tat and Rev. Similarly, a CRISPRa system combined with a CD3-retargeted tBid-encoding vector effectively and selectively targeted latent HIV-infected T ALL cells, inducing their apoptosis.

In conclusion, as a key molecule integrating the intrinsic apoptosis pathway with death domain signaling and the ER stress response, Bid (cBid/tBid) serves as a potent regulator of cell viability. The host utilizes it to maintain homeostasis within the body, while viruses modulate its mitochondrial effects to support their replication, immune evasion, and dissemination strategies. Since apoptosis can either promote or counteract pathogen propagation—depending on the virus, site, stage of infection, or cellular circumstances—Bid may function as a double-edged sword. Its “blade” can be turned against both the host and the pathogen, in accordance with the specific context. Therefore, Bid holds promise as an effective, non-immunogenic asset for highly selective antiviral therapies, as evidenced by preliminary research into its potential in combating HBV-associated HCC and HIV infection.

## Figures and Tables

**Figure 1 ijms-26-02385-f001:**
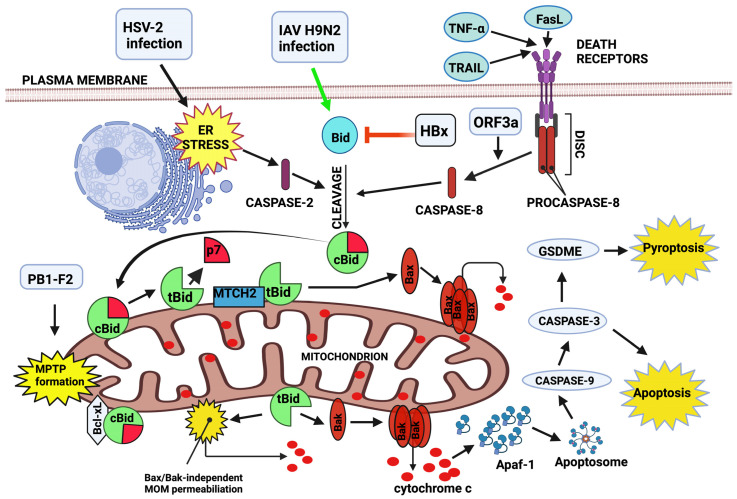
The impact of viral infections on mitochondrial integrity and cell viability: emphasis on Bid regulation and activation. The arrows indicate the regulation of the intracellular level of Bid: green arrows represent upregulation, while red arrows signify downregulation. Abbreviations: Bid (BH3-interacting domain death agonist); TNF-α (tumor necrosis factor); FasL (Fas ligand); TRAIL (TNF-related apoptosis-inducing ligand); DISC (death-inducing signaling complex); tBid (truncated Bid); cBid (complete Bid); ER (endoplasmic reticulum); MTCH2 (mitochondrial carrier homolog 2); MOM (mitochondrial outer membrane); Bax (Bcl-2-associated X); Bak (Bcl-2 homologous antagonist/killer); Apaf-1 (apoptotic protease-activating factor 1); GSDME (gasdermin E); HSV-2 (herpes simplex virus type 2); IAV (influenza A virus). Created in BioRender. Wyżewski, Z. (2025) https://BioRender.com/w33w403 (accessed on 10 January 2025).

**Table 1 ijms-26-02385-t001:** Viral proteins involved in Bid-dependent apoptosis.

Virus	Viral Proteins	Impact on Bid and Bid-Associated Mechanisms	Apoptotic Pathway Impact	References
HBV	HBx	HBx reduces Bid levels	Resistance to Fas- and TNF-α-mediated apoptosis	[107,108]
HSV-2	Misfolded viral proteins	The accumulation of misfolded viral proteins leads to ER stress and the subsequent activation of caspase-2 that cleaves Bid	ER stress-induced intrinsic apoptosis, leading to pyroptosis	[75]
IAV	PB1-F2	PB1-F2 synergizes with tBid in MOM permeabilization	PB1-F2/tBid-mediated intrinsic apoptosis promoting viral replication and immune evasion	[109]
SARS-CoV-2	ORF3a	ORF3a activates caspase-8, leading to Bid cleavage	ORF3a-dependent Bid-mediated apoptosis, contributing to viral propagation and lung injury	[111,112,113,114]

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
