# Peer review of "Bid Protein: A Participant in the Apoptotic Network with Roles in Viral Infections"

_ijms, 2025, doi:10.3390/ijms26062385_

Round 1

Reviewer 1 Report

Comments and Suggestions for Authors

In this paper, the role of Bid protein in apoptosis network and virus infection is reviewed, and the topic selection has important scientific significance and clinical value, which provides a comprehensive theoretical basis for deeply understanding the mechanism of virus infection and developing new treatment strategies. The article is clear in structure and rich in content, covering the structure, function, role in various viral infections and therapeutic potential of Bid protein, and systematically summarizes and deeply analyzes the research in related fields, but there is still some room for improvement.

1. Some mechanisms are not well explained * *: When describing the interaction between Bid protein and other apoptosis regulating molecules, some mechanisms are not well explained, such as the specific details of a series of molecular events triggered by the binding of Bid protein to mitochondrial membrane, and the differences of these events under different virus infection backgrounds, which can be further expanded and refined.

2. Insufficient discussion of research limitations * *: When introducing various research results, the analysis of the limitations of the research itself is not comprehensive enough. For example, when the treatment strategy based on tBid is mentioned, the possible problems, such as the delivery efficiency, potential side effects and stability in complex environment in vivo, are not fully discussed.

3. The use of charts can be optimized * *: Although the article has some schematic diagrams, the number of charts is small and the content is not detailed enough, which fails to fully show the complex molecular mechanism and virus infection process. High-quality charts can be added appropriately, such as drawing more detailed signal path diagrams, to enhance the readability and intuition of the article.

4. Language expression can be simplified * *: The language expression of some paragraphs is complicated and there are repetitive and redundant contents, which affects the efficiency of information transmission to some extent. Some long sentences and complicated paragraphs can be simplified and optimized to make the expression more concise and clear.

5 All the figures are not clear, new figures should be changed!

To sum up, this review has high academic value and research significance, and provides an important reference for the research of Bid protein in the field of virus infection. After the author further improves the above problems, the quality of the article will be significantly improved, which is expected to have a wider impact in this field. It is recommended to publish it after revision.

Author Response

REVIEWER 1:

Ad. 1. Some mechanisms are not well explained * *: When describing the interaction between Bid protein and other apoptosis regulating molecules, some mechanisms are not well explained, such as the specific details of a series of molecular events triggered by the binding of Bid protein to mitochondrial membrane, and the differences of these events under different virus infection backgrounds, which can be further expanded and refined.

Thank you for your comment. The mechanisms triggered by the binding of Bid protein to mitochondrial membrane have been thoroughly explained in the manuscript (lines 161-169), based on studies that focus on the core apoptotic process. Upon tBid-dependent release of cytochrome c from the mitochondria, a series of well-defined molecular events are triggered, which are fundamental to the execution of apoptosis. Cytochrome c binds to apoptotic protease-activating factor 1 (Apaf-1), forming the apoptosome. This complex activates caspase-9, which in turn activates downstream effector caspases, such as caspase-3. Activation of caspases leads to the cleavage of key cellular substrates, resulting in the characteristic morphological and biochemical changes of apoptosis, including chromatin condensation and membrane blebbing. These processes constitute a universal mechanism of apoptosis that is independent of the specific viral context.

Ad. 2. Insufficient discussion of research limitations * *: When introducing various research results, the analysis of the limitations of the research itself is not comprehensive enough. For example, when the treatment strategy based on tBid is mentioned, the possible problems, such as the delivery efficiency, potential side effects and stability in complex environment in vivo, are not fully discussed.

We thank the Reviewer for highlighting the need for a more comprehensive discussion of the limitations of research on tBid-based therapies. In response, we have added a section to address several critical challenges that could arise in translating proposed tBid-based approaches into therapeutic strategies. The title of the added section is: “Challenges and limitations of potential tBid-based therapies” (lines 996-1042). We believe this addition strengthens the manuscript by providing a more balanced perspective on the potential and limitations of tBid-based therapies.

Ad. 3. The use of charts can be optimized * *: Although the article has some schematic diagrams, the number of charts is small and the content is not detailed enough, which fails to fully show the complex molecular mechanism and virus infection process. High-quality charts can be added appropriately, such as drawing more detailed signal path diagrams, to enhance the readability and intuition of the article.

We thank the Reviewer for their suggestion regarding the use of charts to enhance the clarity of the manuscript. While we agree that illustrating the complex molecular mechanisms involved in viral infections and tBid’s role would be beneficial, we have chosen to organize this information in a more systematic way using a table (Table 1, page 9), rather than additional charts or diagrams. This format allows for a clear and concise presentation of key points, and we believe it helps readers better understand the relationships between tBid and various viral infections. We hope this approach fulfills the intention of enhancing the readability and intuitiveness of the article.

Ad. 4. Language expression can be simplified * *: The language expression of some paragraphs is complicated and there are repetitive and redundant contents, which affects the efficiency of information transmission to some extent. Some long sentences and complicated paragraphs can be simplified and optimized to make the expression more concise and clear.

We appreciate the Reviewer’s suggestion to simplify the language expression. In response, we have reviewed the manuscript carefully and made adjustments by removing redundant or repetitive content where appropriate. Additionally, we have restructured some longer sentences to improve readability and clarity. These revisions aim to ensure a more streamlined presentation of the key points.

Ad. 5. All the figures are not clear, new figures should be changed!

The quality of the figure has been improved.

Reviewer 2 Report

Comments and Suggestions for Authors

Dear Authors. Thank you for submitting your review article titled "Bid protein: a participant in the apoptotic network with roles in viral infections.”

This review emphasis the significance of Bid as a pro-apoptotic member of the Bcl-2 family. It is highlights Bid's critical role in regulating apoptosis and elucidates its function as a connector between the extrinsic and intrinsic apoptosis pathways. The insights you provide into how various viruses—including HBV, HSV-2, IAV, and SARS-CoV-2—manipulate Bid signaling underscore the protein's relevance in viral pathogenesis. Rew is  examinates  the mechanisms by which different viruses affect Bid activity, such as the downregulation of Bid by HBV's HBx protein and the enhancement of Bid activity by HSV-2 and IAV. At the end you are suggesting that a deeper understanding of Bid's role in apoptosis and its interactions with viral mechanisms could lead to innovative therapeutic approaches

Comments and Suggestions:

  1. I recommend that you revise the introduction, particularly lines 35 to 47, which currently contain general statements about connectivity in multicellular organisms. Adding more specificity related to the role of apoptosis in viral infections would enhance the introduction. Additionally, providing a clear summary of the pro-apoptotic and anti-apoptotic activities of viral proteins would help contextualize Bid's role and its activation and downregulation within the broader landscape of viral pathogenesis.
  2. Figure 1 Legend: The legend for Figure 1 is somewhat lengthy and would benefit from being succinct. However, the context currently included in the legend could be effectively integrated into Section 3 (lines 349-365) to further illustrate Bid's role in viral infections.
  3. Given your recent publication in 2022 detailing the role of Bcl-xL protein in viral infections within Journal of Molecular Sciences , I encourage you to reference this work discussing the differences in mechanisms of  BCl-xL and Bid related to infection for selected viruses such as IAV, HBV, CoV, and HIV . It would strengthen your review to explain. A brief rationale for your selection of the specific viruses discussed would provide readers with better insight into their significance.
  4. Considering the extensive material you have covered, I recommend including a summary table that outlines the critical proteins involved, their connectivity to Bid, and their interactions with other apoptotic mechanisms for each type of viral infection. This approach would provide a clearer overview for readers, similar to the table featured in the review by Xingchen Zhou et al. (“Virus Infection and Death Receptor-Mediated Apoptosis,” Viruses 2017).
  5. It would be interesting to see an overall summary  on how Bid's function as a connector between extrinsic and intrinsic apoptosis pathways may influence cellular responses to viral infections. This addition should complement the points you are making in lines 968-976.
  6. You propose leveraging Bid's role in apoptosis and its interactions with viral mechanisms for innovative therapeutic strategies. It would be beneficial to briefly discuss current therapeutic approaches and the reasons for their shortcomings. Additionally, it is important to address the challenges associated with modulating such complex mechanisms, which you refer to as a “double-edged sword” (line 973) and have additional discussion about the complexity of this mechanism and potential problems in the context of the whole body/organism. For instance, this is particularly crucial for neurological viruses, where delivering antiviral therapy poses significant challenges and necessitates a thorough understanding of the underlying biological mechanisms.

Overall, your review is comprehensive and provides valuable insights into the role of Bid in the context of viral infections. I look forward to seeing your responses to my suggestions and to the publication of this review in the journal

Author Response

REVIEWER 2

Ad 1.   I recommend that you revise the introduction, particularly lines 35 to 47, which currently contain general statements about connectivity in multicellular organisms. Adding more specificity related to the role of apoptosis in viral infections would enhance the introduction. Additionally, providing a clear summary of the pro-apoptotic and anti-apoptotic activities of viral proteins would help contextualize Bid's role and its activation and downregulation within the broader landscape of viral pathogenesis.

In response to the Reviewer’s suggestion, we have revised the introduction to include more specificity regarding the role of apoptosis in viral infections. We have expanded on how type I programmed cell death (PCD) can serve as part of the host's antiviral strategy by effectively eliminating infected cells, disrupting pathogen replication, and preventing the spread of the virus throughout the organism. Furthermore, we elaborated on the impact of different viruses on apoptosis, highlighting how they can either promote or suppress cell death through the modulation of molecular signaling pathways. We also included a more detailed explanation of the involvement of both extrinsic and intrinsic apoptosis pathways in viral infections. These additions aim to provide a clearer context for the role of apoptosis in viral pathogenesis and to set the stage for the discussion of Bid’s role in this process.

In lines 50-61, we have added the following fragment:

“Type I PCD can be part of the host's antiviral strategy. By effectively eliminating infected cells, it may disrupt pathogen DNA/RNA replication and the production of progeny virions. On a systemic scale, it can prevent the spread of the "intruder" throughout the organism. However, distinct viruses can affect the viability of infected cells, promoting pathogen propagation, impairing the immune response, or establishing a latent stage that allows them to persist in the host while evading its defense mechanisms. Viral infections can either promote or suppress apoptosis, as viruses encode proteins that interfere with molecular signaling. The modulation of cell death may involve both the extrinsic and intrinsic pathways of type I PCD.

The BH3-interacting domain death agonist (Bid), a protein that bridges receptor-mediated and mitochondria-dependent apoptotic cascades, may play a role in various viral infections”.

Additionally, in line 44, we have added the formulation “and infected” to the sentence: “The old, damaged, and malfunctioning ones undergo elimination via type I programmed cell death (PCD), also called apoptosis,” to underline the role of apoptosis in counteracting viral infections. These additions aim to provide a clearer context for the role of apoptosis in viral pathogenesis and to set the stage for the discussion of Bid’s role in this process.

Ad. 2. Figure 1 Legend: The legend for Figure 1 is somewhat lengthy and would benefit from being succinct. However, the context currently included in the legend could be effectively integrated into Section 3 (lines 349-365) to further illustrate Bid's role in viral infections.

We have added the following fragment to the section 3 (lines 398-424): “According to the illustration, Bid links the extrinsic and intrinsic apoptosis pathways. Stimulation of death receptors by their ligands, such as TNF-α, Fas ligand (FasL), and TRAIL, induces the formation of DISC and the activation of caspase-8. This enzyme activates Bid by cleaving it into two subunits: tBid and p7. They remain non-covalently bound to form cBid. Bid can also be cleaved by caspase-2, activated in response to the ER stress. cBid migrates to the mitochondria and releases the p7 subunit. MTCH2 interacts with tBid, promoting its anchoring in the MOM and facilitating its conversion into active, functional conformation. tBid stimulates Bax and/or Bak proteins to form oligomeric pores in the MOM, allowing the release of apoptogenic factors from the mitochondrial intermembrane space into the cytosol. One of them, cytochrome c, binds to apoptotic protease-activating factor 1 (Apaf-1) to form heterodimers, which then assemble into a heptameric complex known as the apoptosome. The subsequent events include the activation of caspases-9 and -3, leading to apoptosis or, in certain cases, to gasdermin E (GSDME)-dependent pyroptosis. Additionally, cBid can bind to Bcl-xL, reducing its inhibition of Bax, thereby promoting mitochondrial dysfunction. tBid may also cause Bax/Bak-independent MOM permeabilization. Various viruses, including HBV, HSV-2, SARS-CoV-2, and IAV, can impact cell viability by modulating the Bid-mediated apoptosis pathway. While HSV-2, IAV, and SARS-CoV-2 promote the Bid (cBid/tBid)-mediated apoptotic cascade, HBV may prevent it. Pathogens can trigger the activation of caspases acting upstream of Bid proteolysis. HSV-2 infection leads to ER stress and subsequent caspase-2-catalyzed Bid cleavage. Meanwhile, the activation of caspase-8 can be mediated by open reading frame 3a (ORF3a), the proapoptotic protein of SARS-CoV-2. Polymerase basic 1 frame 2 (PB1-F2), a product of IAV, may cooperate with adenine nucleotide translocase 3 (ANT3) and voltage-dependent anion channel 1 (VDAC1) to form the mitochondrial permeability transition pore (MPTP), enhancing tBid’s effect on MOM integrity. IAV H9N2 infection upregulates the intracellular level of Bid, whereas the HBV X protein (HBx) exerts the opposite effect on the abundance of this Bcl-2 family member.”

In addition, we have shortened the caption below Figure 1 by replacing the detailed description with the following concise text:

“The impact of viral infections on mitochondrial integrity and cell viability: emphasis on Bid regulation and activation. The arrows indicate the regulation of the intracellular level of Bid: green arrows represent upregulation, while red arrows signify downregulation.

Abbreviations: Bid (BH3-interacting domain death agonist); TNF-α (tumor necrosis factor); FasL (Fas ligand); TRAIL (TNF-related apoptosis-inducing ligand); DISC (death-inducing signaling complex); tBid (truncated Bid); cBid (complete Bid); ER (endoplasmic reticulum); MTCH2 (mitochondrial carrier homolog 2); MOM (mitochondrial outer membrane); Bax (Bcl-2-associated X); Bak (Bcl-2 homologous antagonist/killer); Apaf-1 (apoptotic protease-activating factor 1); GSDME (gasdermin E); HBV (hepatitis B virus); HSV-2 (herpes simplex virus type 2); SARS-CoV-2 (severe acute respiratory syndrome coronavirus 2); IAV (influenza A virus).”.

Ad. 3. Given your recent publication in 2022 detailing the role of Bcl-xL protein in viral infections within Journal of Molecular Sciences , I encourage you to reference this work discussing the differences in mechanisms of  BCl-xL and Bid related to infection for selected viruses such as IAV, HBV, CoV, and HIV . It would strengthen your review to explain. A brief rationale for your selection of the specific viruses discussed would provide readers with better insight into their significance.

In response to the Reviewer's suggestion, we have expanded the manuscript to further discuss the role of Bcl-xL in viral infections. Additional paragraphs have been incorporated into the relevant sections, highlighting the involvement of Bcl-xL in virus-cell interactions and its potential impact on apoptosis regulation:

1) “Interestingly, apart from Bid, another member of the Bcl-2 protein family, Bcl-xL, also contributes to the survival of HBV-associated HCC cells. Our previous review [24] described the importance of Bcl-xL in the context of virus-cell interactions. In HBV-associated HCC, an increase in the intracellular level of Bcl-xL has been reported, which, together with the loss of Bid, may contribute to the antiapoptotic effect observed in these cells” (lines 481-486).

2) “In addition to the proapoptotic activity of PB1-F2 and tBid in IAV-infected cells, antiapoptotic representatives of the Bcl-2 family, including Bcl-xL, may participate in the regulation of cell viability in the course of viral replication. Research suggests that Bcl-xL, together with Bcl-2 and Bcl-w, counteracts MOM permeabilization at the early stage of infection, preventing premature apoptosis and promoting viral replication. Notably, Bcl-xL may also play a role in the later stages of infection. Then, the upregulation of this protein via the Janus kinase/signal transducers and the activators of transcription (JAK-STAT) pathway has been linked to a switch from apoptosis to pyroptosis, an inflammatory form of cell death that favors viral clearance. These findings highlight the dynamic balance between pro- and antiapoptotic Bcl-2 family members in the progression of IAV infection.” (lines 649-459).

3) In addition to Mcl-1, another Bcl-2 family member important for the pathogenesis of SARS is Bcl-xL. Research suggests that the E protein of SARS-CoV-2 may bind to Bcl-xL, reducing host cell viability and potentially contributing to virus-associated lymphopenia. (lines 740-743).

4) The role of Bcl-xL in HIV infection was previously mentioned in this fragment (lines 919-927): “In vitro studies revealed that during the maturation of HIV-infected monocytes into macrophages, the viral envelope glycoproteins (gp120/gp41) upregulate the production of macrophage colony-stimulating factor (M-CSF), a cytokine that is involved in the increase of intracellular levels of Bcl-2 family prosurvival members, including Mcl-1, Bcl-xL, and Bcl2-related protein A1 (Bcl2-A1). This upregulation results in significant desensitization of infected cells to external apoptotic stimuli, such as TRAIL. In light of the obtained results, HIV appears to indirectly modulate the apoptosis molecular network by suppressing its mitochondria-dependent branch, which helps maintain the viral reservoir”

A brief rationale for the selection of the specific viruses was presented in the following fragment added to the introduction (lines 71-77): “The pathogens selected for this review pose a significant challenge to modern medicine. They infect humans, resulting in a wide range of health consequences, from chronic diseases to severe acute outcomes. Due to their clinical significance and documented association with Bid-dependent apoptosis regulation, these pathogens were chosen for this article. A deeper understanding of Bid's role in determining the fate of infected cells could aid in the search for effective antiviral treatment strategies”.

Ad. 4.  Considering the extensive material you have covered, I recommend including a summary table that outlines the critical proteins involved, their connectivity to Bid, and their interactions with other apoptotic mechanisms for each type of viral infection. This approach would provide a clearer overview for readers, similar to the table featured in the review by Xingchen Zhou et al. (“Virus Infection and Death Receptor-Mediated Apoptosis,” Viruses 2017).

The table summarizing the impact of critical viral proteins on Bid and Bid-dependent apoptotic mechanisms has been added (Table 1, page 9).

Ad. 5.  It would be interesting to see an overall summary on how Bid's function as a connector between extrinsic and intrinsic apoptosis pathways may influence cellular responses to viral infections. This addition should complement the points you are making in lines 968-976.

 The overall summary was added (lines 1061-1070):

“Bid plays a pivotal role as a molecular linker that integrates two apoptotic pathways: the extrinsic and intrinsic. The intersection of these signaling cascades is crucial for maintaining cellular homeostasis and responding to stress stimuli. Bid is involved in regulating the balance between cell death and survival, adjusting this equilibrium to the varying needs of the multicellular organism. However, viruses can influence cell viability in ways that favor the development of infection, promoting viral propagation, immune evasion, and pathogenesis, with consequences such as tissue damage and oncogenesis. By modulating Bid levels and activity, either synergizing with or antagonizing it, viruses enhance or disrupt the bridge between these two apoptosis pathways. This alters the holistic apoptotic signaling network in a manner that benefits the virus while harming the host”.

Ad. 6. You propose leveraging Bid's role in apoptosis and its interactions with viral mechanisms for innovative therapeutic strategies. It would be beneficial to briefly discuss current therapeutic approaches and the reasons for their shortcomings. Additionally, it is important to address the challenges associated with modulating such complex mechanisms, which you refer to as a “double-edged sword” (line 973) and have additional discussion about the complexity of this mechanism and potential problems in the context of the whole body/organism. For instance, this is particularly crucial for neurological viruses, where delivering antiviral therapy poses significant challenges and necessitates a thorough understanding of the underlying biological mechanisms.

We have added a brief discussion on current therapeutic approaches in HCC treatment (lines 792-816):Treatment of HCC can be either curative or palliative, depending on the stage of tumor development and prognosis. As HCC is resistant to chemotherapy, medical interventions include invasive surgical methods, such as partial or major hepatic resection and liver transplantation. These approaches offer hope for survival at early stages of tumor development; however, HCC is often diagnosed at more advanced stages, in which treatment options are more limited. Furthermore, liver transplantation presents the challenge of finding a suitable donor, while partial hepatic resection carries the risk of recurrence and peri- or postoperative mortality due to an insufficient mass of the remaining liver for proper regeneration.

Alternative curative methods for HCC therapy include chemical and physical ablations, such as ethanol or radiofrequency ablation. These approaches have the advantage of reducing systemic side effects; however, a significant limitation is the higher recurrence rate compared to resection. The condition of the liver, the stage at diagnosis, the size of the tumor, the number of foci, and the metastatic potential of the neoplasm are key factors that determine whether a patient is classified for curative or palliative treatment. Due to poor prognosis, the majority of individuals with HCC fall into the latter category. They may undergo treatment through embolization, systemic chemotherapy, and/or molecular targeting. The first method is based on obstructing the blood supply to the tumor tissue and may be supported by cytostatic drugs such as doxorubicin and cisplatin, or combined with radiotherapy. However, systemic chemotherapy and radiation therapy show relatively low efficacy in treating HCC and are associated with a wide range of side effects. Molecular targeting using antitumorigenic and angiogenic agents such as sunitinib, linifanib, brivanib, and sorafenib faces several challenges, including limited efficacy, low effectiveness against metastatic tumors, high treatment costs, liver toxicity, and numerous side effects”.

Regarding the limitations of current HIV treatment, we have incorporated the following discussion (lines 963-972):Current anti-HIV treatment strategies are based on antiretroviral therapy (ART), which includes nucleotide reverse transcriptase inhibitors (NRTIs), protease inhibitors (PIs), integrase strand transfer inhibitors (INSTIs), entry inhibitors (EIs), and C-C chemokine receptor type 5 (CCR5) antagonists. The advantages of ART include significant elongation of the patient’s lifespan, reduction in viremia, strengthening of the host immune system, and an increase in the number of CD4+ lymphocytes, which diminishes the risk of AIDS development. However, as mentioned earlier, the pathogen is not fully eliminated and remains hidden in cells. Additional limitations and disadvantages of ART include side effects such as gastrointestinal issues, liver and/or kidney failure, hypercholesterolemia, and an elevated susceptibility to cardiovascular diseases”.

We thank the Reviewer for pointing out the complexity of modulating Bid-based signaling and its “double-edged sword” nature. We have expanded the discussion to address this important aspect in the revised version of the manuscript. We have added the subsection addressing this issue: “Challenges and limitations of potential tBid-based therapies” (lines 996-1042).